# Predicting synthetic mRNA stability using massively parallel kinetic measurements, biophysical modeling, and machine learning

Daniel P. Cetnar[1], Ayaan Hossain [2], Grace E. Vezeau[3] & Howard M. Salis [1,3,4] ✉

mRNA degradation is a central process that affects all gene expression levels, though it remains challenging to predict the stability of a mRNA from its sequence, due to the many coupled interactions that control degradation rate. Here, we carried out massively parallel kinetic decay measurements on over 50,000 bacterial mRNAs, using a learn-by-design approach to develop and validate a predictive sequence-to-function model of mRNA stability. mRNAs were designed to systematically vary translation rates, secondary structures, sequence compositions, G-quadruplexes, i-motifs, and RppH activity, resulting in mRNA half-lives from about 20 seconds to 20 minutes. We combined biophysical models and machine learning to develop steady-state and kinetic decay models of mRNA stability with high accuracy and generalizability, utilizing transcription rate models to identify mRNA isoforms and translation rate models to calculate ribosome protection. Overall, the developed model quantifies the key interactions that collectively control mRNA stability in bacterial operons and predicts how changing mRNA sequence alters mRNA stability, which is important when studying and engineering bacterial genetic systems.

Engineering high-performance genetic systems requires the development of more comprehensive design platforms that can better predict how genetic sequence controls gene expression levels. While there are sequence-to-function models that are commonly used to predict bacterial transcription initiation rates and translation initiation rates[1–3], there have been comparably fewer efforts to develop models that predict bacterial mRNA degradation rates[4–6]. mRNA degradation is an important contributor to genetic system function as even small changes to mRNA sequence can alter mRNA decay rates by 10-fold[4,5,7–9], which affects both steady-state protein expression levels and "turn off" times in response to gene regulatory dynamics. By enabling the rational engineering of genetic systems and filling in a key gap in existing design platforms, the development of a predictive model of mRNA decay will have broad applications in the engineering of metabolic pathways, genetic circuits, biosensors, and genomes[10–30].

A key challenge to predicting mRNA decay rates is that degradation is controlled by several coupled interactions, involving ribosome protection, mRNA structure, G-quadruplexes, RNase binding sites, RNA modifying enzymes, and other RNA-binding proteins[31–41]. An ideal model should accurately predict mRNA decay rates across diverse mRNA sequences with combinations of such interactions while pinpointing the sequence determinants that are responsible for a particular mRNA's decay rate. Here, we overcame this challenge by designing, constructing, and characterizing 62,120 synthetic mRNAs with combinations of interactions with varied strengths, carrying out kinetic mRNA level measurements on rifampicin-treated cells to quantify their mRNA decay rates. Leveraging this dataset, we then combined biophysical modeling and machine learning to develop a sequence-to-function model of mRNA degradation rate. Overall, the model accurately predicts steady-state mRNA levels and mRNA half-

[1]Department of Chemical Engineering, The Pennsylvania State University, University Park, PA, USA. [2]Graduate Program in Bioinformatics and Genomics, The Pennsylvania State University, University Park, PA, USA. [3]Department of Biological Engineering, The Pennsylvania State University, University Park, PA, USA. [4]Department of Biomedical Engineering, The Pennsylvania State University, University Park, PA, USA. ✉e-mail: salis@psu.edu

lives, while identifying key sequence determinants in the 5′ untranslated region (UTR) that have large effects on the mRNA degradation process.

Our model-building efforts relied on extensive past work identifying the enzymes responsible for mRNA degradation, including the endonucleases RNase E[32,33], RNase G[32,34], and RNase III[35,36]; the exonucleases RNase II, RNase R, and PNPase[37,38]; RNA modifying enzymes, such as RppH[39,40] and Poly(A) polymerase[37]; and other RNA-binding proteins that alter mRNA translation rates and may protect mRNAs from RNase activity (e.g., CsrA[42] and Hfq[43]). These proteins have distinct mRNA sequence and structural binding patterns with a high level of degeneracy, and are often loosely associated with a multi-protein complex−the RNA degradosome−that processively degrades mRNA in a coordinated multi-step process[44]. We also built on our prior work, where we found that the dephosphorylation activity of RppH−a key preceding step to end-dependent mRNA decay−strongly depended on the first few nucleotides of the mRNA transcript, though only 16 RppH binding sites were characterized[4]. The mRNA's translation initiation rate also played an outsized role in controlling its decay rate through varying the density of protective ribosomes bound to the mRNA[4,45]. Finally, increasing the amount of unstructured mRNA in the 5′ UTR accelerated mRNA decay by providing more binding sites for RNase E/G, which catalyze key rate-limiting steps in the mRNA decay process in *Escherichia coli*[4]. However, prior efforts to measure the effects of these interactions have relied on characterization of natural mRNAs or small sets of synthetic mRNAs[46–48]. These datasets were insufficient to build a predictive sequence-to-function model that is both accurate and generalized, particularly due to a paucity of characterized mRNAs with combinations of systematically varied interactions.

## Results

### Massively parallel design and construction of 5′ UTRs to vary mRNA decay rate

We designed 62,120 5′ UTRs using combinations of sequence determinants that are proposed to alter mRNA decay rates, including: (1) the RppH binding site, which includes the first four nucleotides of the mRNA transcript; (2) the length and sequence composition of single-stranded (unstructured) regions in the 5′ UTR; (3) the sizes and folding energetics of mRNA secondary structures, varying the number of mRNA hairpins, bulges, internal loops within the 5′ UTR; (4) the presence of mRNA tertiary structures, including G-quadruplexes and i-motifs; and (5) designed ribosome binding site sequences that systematically varied the translation rate of a protein coding sequence (CDS).

Each designed mRNA transcript contains combinations of these factors to broadly explore the sequence-structure-function space, collected together into design groups. For example, in one design group, we designed 1280 5′ UTR sequences where each sequence varied the RppH binding site sequence (256 variants), followed by a single-stranded region (a 16-nt polyA motif) and a different ribosome binding site sequence (5 variants) that varied the CDS translation rates across a 10,000-fold range. We created four more design groups (1116 to 1772 designed sequences each) that systematically varied the RppH binding site and CDS translation rate, while varying the sequence and composition of the single-stranded RNA (ssRNA) region using polyU, AAACAAA, AAAGAAA, or UUUGUUU motifs. We created six more design groups (1170 to 8338 designed sequences each) that systematically varied the sequence and structure of the upstream 5′ UTR region, either utilizing polyA, polyG, polyC, or polyU tracts of varying length, shortening the region to 6-nt, inserting mRNA hairpins with varying duplex lengths (2, 8, or 16 duplexed base pairings), or removing the upstream region completely, all while varying the RppH binding site and CDS translation rates in each design group.

The next several design groups tested the importance of specific types of mRNA structures or binding sites. We began with designing 8073 5′ UTR sequences that have highly diverse sequences, but where

subsets of sequences fold into a targeted secondary structure, using our recently developed Non-Repetitive Parts Calculator with an RNA structural constraint to carry out rational design[49]. The purpose of this design group is to distinguish between sequence motifs that specifically control RNase binding versus structural motifs that control overall accessibility to RNase binding sites. We next designed 3111 5′ UTR sequences with highly non-repetitive G-quadruplexes, using the pattern [(3G-6N)$_4$], to test whether these compact tertiary structures can block RNase binding. Similarly, we designed 3310 5′ UTR sequences with i-motifs structures, using the pattern [(3C-6N)$_4$], to test whether these non-canonical structures utilizing C-rich Hoogsteen base pairing can block RNase binding. We designed 1298 5′ UTR sequences containing between 1 and 5 sequence motifs previously suggested to bind to RNase E[50] or that bind to CsrA[51], which is a factor that regulates translation rates and may protect mRNAs from degradation. Finally, we designed 974 5′ UTR sequences that systematically varied CDS translation rates, using the RBS Calculator v2.1 to design ribosome binding sites with varied standby sites, Shine-Dalgarno sequences, and inhibitory mRNA structures[2,52].

Each 5′ UTR was encoded within a 170-nt oligonucleotide that is paired to a distinct DNA barcode sequence. All oligonucleotides contain flanking PCR primer binding sites and two pairs of restriction sites to support 2-step library-based cloning. Oligonucleotides were synthesized in five separates oligopools (Genscript). Each oligopool was then PCR amplified, digested, and ligated into a plasmid-based expression system, followed by a second round of digestion and ligation to insert a constant sfGFP coding sequence (Methods). The final library of barcoded genetic systems uses a constant bacterial promoter (J23100) and the designed 5′ UTR to express the sfGFP reporter. However, we do not use the sfGFP reporter for measuring protein levels; instead, its purpose is to create a nominal expression environment where ribosomes bind to the mRNA transcript and provide protection against RNases, depending on its translation rate (Fig. 1A). During the 2-step cloning, the barcode sequence is repositioned downstream of the sfGFP reporter to minimize its effects on the mRNA decay rate. After separate cloning and transformation of the five plasmid libraries into *E. coli* DH5α cells, we combined cells together into a single-cell library and carried out MiSeq next-generation sequencing to assess library coverage. Overall, MiSeq sequencing yielded 7,955,477 reads and identified 59,721 expected barcode sequences (96.1% library coverage).

### Massively parallel kinetic measurements to characterize mRNA decay rates

We then carried out massively parallel mRNA level measurements to determine the steady-state and kinetic decay rates of each engineered mRNA transcript (Methods). Mixed cell libraries were serially cultured in selective LB media at 37 °C so that all cells were maintained in the exponential growth phase for at least 4 h. At the starting timepoint (T0), cell samples were taken, followed by addition of rifampicin. Rifampicin directly binds to *E. coli* RNA polymerase and halts transcription of new mRNA, but does not otherwise affect the mRNA decay machinery[53]. Cells samples were then taken 2, 4, 8, and 16 min after addition of rifampicin, while cells continued to grow. All cell samples were protected from chemical degradation using RNAprotect reagent (Qiagen). Total RNA was immediately extracted from all timepoints. Plasmid DNA was extracted from T0 cell samples, followed by barcoded amplicon generation. A known and constant amount of spike-in RNA was then added to each total RNA sample for cross-timepoint normalization, followed by rRNA depletion, cDNA synthesis, and barcoded amplicon generation. Deep sequencing was then carried out on the barcoded amplicon libraries (Illumina NovaSeq), yielding 1.6 billion reads mapped to expected RNA barcodes or spike-in-control RNA and another 301 million reads mapped to expected DNA barcodes. As expected from mRNA decay, the number of reads matching mRNA-

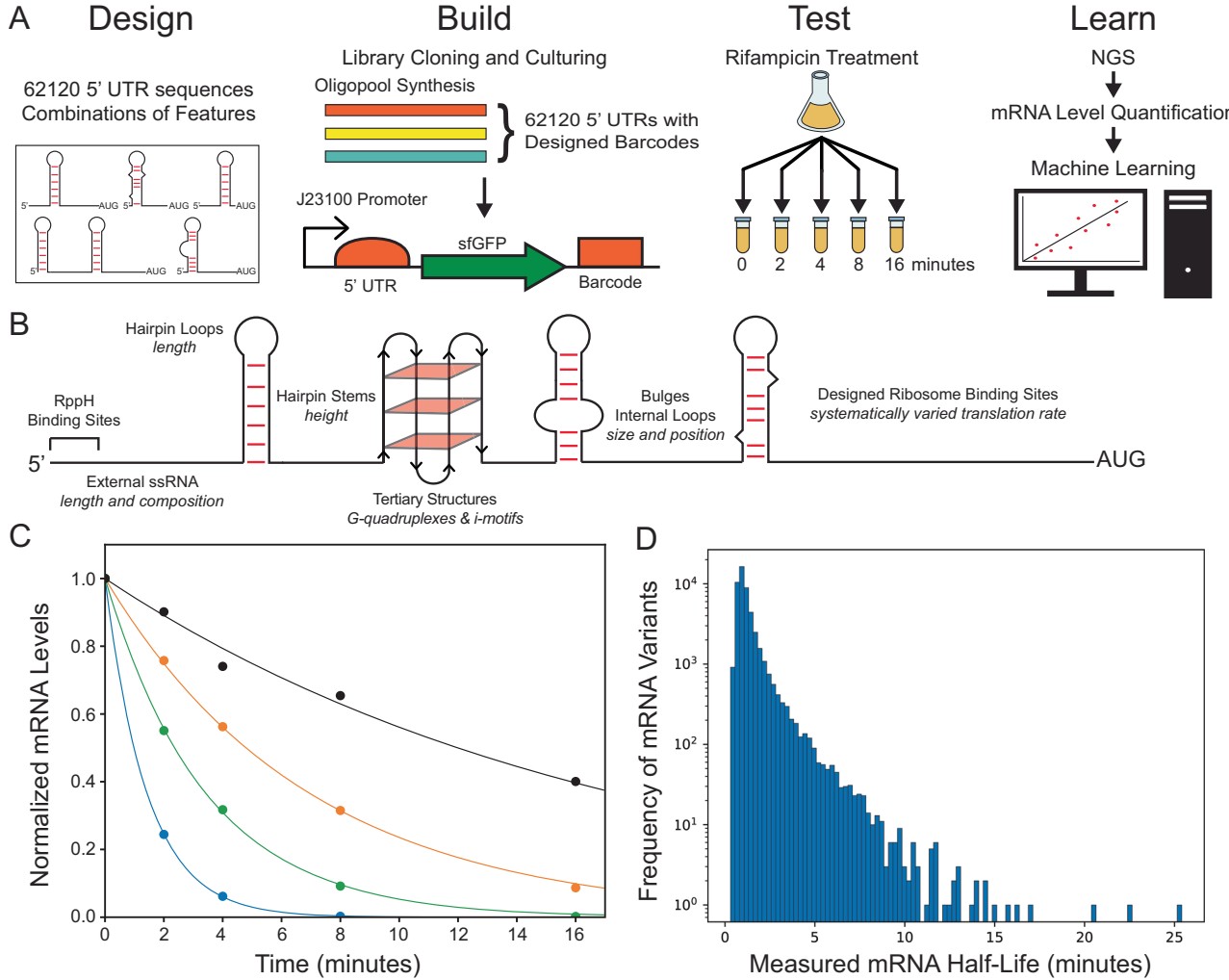

**Fig. 1 | Massively parallel design and kinetic decay measurements. A** 62,120 barcoded genetic systems with varied 5′ UTR sequences were constructed using oligopool synthesis and library-based cloning. mRNA stabilities in exponentially growing *E. coli* cells were measured using rifampicin treatment to halt transcription initiation, followed by kinetic mRNA level measurements using DNA-Seq and RNA-Seq. **B** mRNA sequences were designed with combinations of factors affecting mRNA decay rate, including RppH binding affinity, mRNA secondary & tertiary structure, and mRNA translation rate. **C** Time course mRNA level measurements (dots) were fitted to exponential decay functions (lines), using spike-in RNA controls for normalization. Four characterized mRNAs with widely different mRNA decay rates are shown. **D** The distribution of measured half-lives are shown.

derived barcodes steadily decreased across timepoints with a corresponding increase in reads matching the spike-in control RNA (Supplementary Table 1). We then counted the number of each DNA and mRNA barcode that mapped to reads. DNA barcode counts quantify the composition of the cell library at the T0 timepoint, while mRNA barcode counts quantify mRNA levels at the T0 to T16 timepoints. We found that 56,816 DNA barcodes (91.5%) and 58,080 mRNA barcodes (93.5%) had at least 100 mapped reads at the T0 timepoint, showing that deep sequencing yielded enough reads to broadly cover the cell library. The median read counts were 1576 and 2015, respectively. Cell library coverage remained high after rifampicin treatment with between 82.2 and 91.4% of mapped mRNA barcodes having at least 100 reads across each timepoint. All genetic part sequences, designed 5′ UTR variants, barcode sequences, and barcode counts are available in Supplementary Data 1.

For each 5′ UTR variant, we then quantified its mRNA levels over each timepoint and assessed how well the kinetic mRNA levels match to an exponential decay curve. We quantified mRNA levels by dividing the mRNA barcode counts at each timepoint by the DNA barcode counts at the T0 timepoint. We then compared the kinetic mRNA levels to the exponential curve $M(t) = M_0 \, e^{-kt}$, where $M_0$ is the mRNA level at

the T0 timepoint, $t$ is time (minutes), $M(t)$ is the mRNA level at each timepoint, and $k$ is the first-order kinetic decay constant (1/min). mRNA half-lives are calculated according to $t_{1/2} = \log(2)/k$, where the natural log is used. In Fig. 1C, we show measured mRNA levels at each timepoint in comparison to fitted exponential decay curves, selecting mRNAs with widely different decay kinetics. Overall, we found that 50,048 mRNAs (80.6%) followed exponential decay kinetics ($R^2 > 0.75$ for each curve) with in vivo mRNA half-lives that varied from 0.31 to 25.4 min (Fig. 1D). These results show that the designed 5′ UTR sequences greatly altered in vivo mRNA decay rates with half-lives that span the physiologically relevant range from seconds to minutes[54].

**Multi-factor sequence determinants controlling mRNA decay**
Our first data analysis step was to determine how the sequence design factors affected the measured mRNA decay rates. Initially, we compared how changing a single design factor altered mRNA decay rates. However, we found that focusing on only a single design factor resulted in larger intra-category differences than inter-category differences, indicating that multiple factors are collectively controlling the mRNA's decay rate with equally important contributions (Supplementary Information). For example, after tallying the decay rates for all

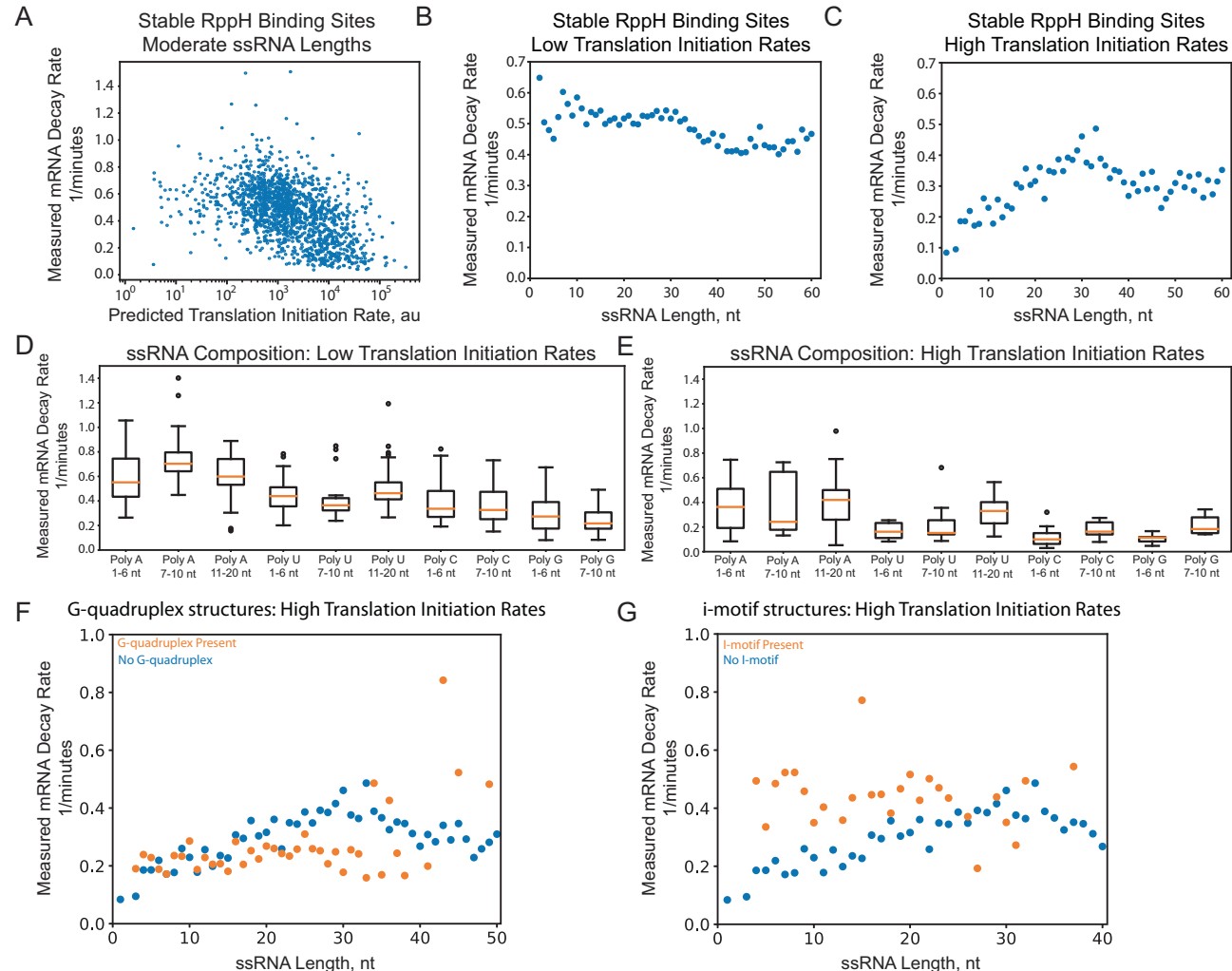

**Fig. 2 | Multi-factor sequence determinants controlling bacterial mRNA decay.**
**A** Measured mRNA decay rates are shown when varying the predicted mRNA
translation initiation rates, while only including 5′ UTRs with stable (less active)
RppH binding sites and moderate single-stranded RNA lengths (Pearson $R = -0.31$,
two-tailed $p = 4.2 \times 10^{-27}$). **B** Measured mRNA decay rates are shown when varying
single-stranded RNA lengths inside the 5′ UTR, while only including 5′ UTRs with
stable RppH binding sites and low predicted translation initiation rates (<5000 au)
(small size effect = 0.0008 for 0 to 30 nt, Pearson $R = -0.19$, two-tailed
$p = 1.6 \times 10^{-14}$). **C** Measured mRNA decay rates are shown when varying single-
stranded RNA lengths inside the 5′ UTR, now including 5′ UTRs with stable RppH
binding sites and high predicted translation initiation rates (>5000 au) (large size

effect = 0.01 for 0 to 30 nt, $R = 0.924$, two-tailed $p = 1.3 \times 10^{-14}$). Measured mRNA
decay rates are shown when varying the length and composition of single-stranded
RNA regions inside 5′ UTRs, using ribosome binding sites with either (**D**) low or (**E**)
high predicted translation initiation rates. Box plots show the median value of
measured mRNA decay rates ($N > 100$ mRNAs) across each category (orange line),
the 25% and 75% quartile boundaries (boxes), the maximum and minimum of the in-
distribution data (bars), and the outliers (points). Measured mRNA decay rates are
shown when introducing either (**F**) G-quadruplex tertiary structures or (**G**) i-motif
tertiary structures into highly translated mRNAs as compared to highly translated
mRNAs lacking each type of structure, each indexed by the amount of single-
stranded RNA in their 5′ UTRs.

mRNAs starting with the same RppH binding site and calculating their
mean and quartile values, we found that the differences in the mean
decay rates across different RppH binding sites were much smaller
than the differences in the upper and lower quartiles when using the
same RppH binding site (Supplementary Fig. 1). We found similar
outcomes when using single design factor analysis to determine the
effect of changing the CDS' translation rates (Supplementary Fig. 2) or
the amount of ssRNA in the 5′ UTR (Supplementary Fig. 3). This initial
analysis motivated the use of multi-factor categorization to determine
how each factor controlled mRNA decay rates, while keeping other
factors constant.

We next applied multi-factor categorization to determine how
individual design factors controlled mRNA decay rates. We collected
together characterized mRNAs that had RppH binding sites that were
more likely to confer greater mRNA stability (20 most stable RppH
binding sites) as well as characterized mRNAs that had either high or

low predicted CDS translation rates (higher or lower than 5000 au on
the RBS Calculator v2.1 scale). We also collected together mRNAs that
had moderate amounts of ssRNA in their 5′ UTR (11 to 40 nt). Notably,
when we focus on mRNAs with stable RppH binding sites and mod-
erate ssRNA lengths, we find an inverse relationship between the CDSs'
predicted translation initiation rates and their measured decay rates
(Pearson $R = -0.31$, $p$-value $= 4 \times 10^{-27}$, $N = 1527$ mRNAs) with higher
translation rates providing mRNAs with more stability (Fig. 2A), con-
firming that some design factors may only have a clear effect on mRNA
decay rates when other contributing factors are kept relatively
constant.

In another example, we found that increasing the amount of
ssRNA in the 5′ UTR increased the mRNAs' decay rates, but the rela-
tionship only becomes clear when the analysis focused on mRNAs with
both stable RppH binding sites and high predicted translation initia-
tion rates (Fig. 2B, C). When mRNAs had low predicted translation

rates, their measured mRNA decay rates were already high and any additional ssRNA in their 5′ UTR did not significantly increase the decay rate (Fig. 2B). The sequence composition of the ssRNA also contributed to changes in mRNA decay (Fig. 2D, E) independent of their effect on the mRNA's structure and translation rate, possibly due to changes in the mRNAs' rigidity and persistence length, which control its overall accessibility to RNase binding. PolyA motifs appeared to have a larger impact on mRNA decay than other homopolymer motifs, particularly when mRNAs had high predicted translation rates (Fig. 2E), based on both the larger changes in the median mRNA decay rates and the greater spread in their distributions.

Finally, we found that G-quadruplex tertiary structures had a protective effect on mRNA stability, generally lowering their decay rates, while i-motif tertiary structures had no apparent protective effect on mRNA stability (Fig. 2G, F). In this multi-factor analysis, we compared mRNAs that form tertiary mRNA structures versus mRNAs that only fold into secondary mRNA structures, focusing only on mRNAs with high translation initiation rates that should have a low baseline mRNA decay rate (high stability). In both cases, we calculate the amount of ssRNA in the 5′ UTR to carry out equal-category (apples-to-apples) comparisons; for mRNAs designed with G-quadruplexes and i-motif tertiary structures, we calculate the amount of ssRNA when only secondary mRNA structures are allowed to form. In this way, our comparative analysis determines the contribution of the tertiary mRNA structure itself, independent of other contributing factors. We found that the introduction of G-quadruplexes into the 5′ UTR exhibited low mRNA decay rates regardless of the secondary mRNA structures that would otherwise form with varied amounts of ssRNA (Fig. 2F). In the absence of a G-quadruplex, a longer ssRNA region caused the mRNA decay rate to increase by about 0.01 1/min per ssRNA nucleotide up to a length of about 33 nucleotides (linear regression; $R = 0.926$, $p = 5.14 \times 10^{-15}$, $N = 31$) with a peak mRNA decay rate of about 0.4 1/min. But when a G-quadruplex was introduced into the 5′ UTR, the mRNA decay rate was generally constant at about 0.2 1/min and did not depend on the 5′ UTR length (for up to 33 nucleotides of added sequence). In contrast, using the same analysis, we found that the introduction of i-motifs generally increased mRNA decay rates, compared to an equivalent mRNA that only formed secondary mRNA structures (Fig. 2G). For example, introducing an i-motif into a shorter 5′ UTR caused its mRNA decay rate to be high (about 0.5 1/min) in comparison to mRNAs with an equivalent length of ssRNA (5 to 10 nt), which had mRNA decay rates of about 0.2 1/min.

## Development of a predictive model using biophysics and machine learning

We next developed a quantitative sequence-to-function model that predicts a bacterial mRNA's decay rate, combining both biophysical model calculations and machine learning to quantify the most important design factors that control mRNA stability (Fig. 3). Our first step was to generate a long list of candidate sequence, structural, and functional determinants that may contribute to mRNA levels and mRNA decay rate, including the promoter's transcription initiation rate, the RppH binding sites, the types and locations of mRNA structures, the amount of ssRNA in the 5′ UTR, the sequence compositions of different regions, and the translation initiation rate of the CDS. We applied several biophysical models to calculate these determinants, using only the genetic system's sequence as the input. For example, we used a newly developed thermodynamic model (the Promoter Calculator v1.0) to calculate the promoters' site-specific transcription initiation rates; even though the same promoter (J23100) is used to transcribe all mRNAs, its transcriptional start site and transcription initiation rate can change based on the first 20 nucleotides of the mRNA transcript, called the initial transcribed region (ITR)[1]. We used another thermodynamic model (the RBS Calculator v2.1) to calculate the translation initiation rates of the CDSs, which vary greatly across different 5′ UTR sequences[2]. We also

applied RNA folding algorithms (Vienna RNA v2.4.11) to calculate the mRNA structures that form in each 5′ UTR-CDS region, including the types and locations of the thermodynamically most stable secondary structures and G-quadruplexes[55]. We used these structural calculations to tally several potential determinants, including hairpin duplex lengths, hairpin loop lengths, the number of bulges and internal loops inside hairpins, the amount of ssRNA in between hairpins, and the sequence composition of these ssRNA regions.

Based on our prior data analysis, we found that the first four nucleotides of the mRNA transcript (the RppH binding site) have a particularly outsized impact on a mRNA's decay rate. From that, we recognized that small differences in the promoter's transcriptional start site will generate distinct mRNA isoforms with different RppH binding sites. Notably, a key advantage of the Promoter Calculator model is its ability to predict the transcription initiation rate for each potential start site. We accordingly applied the Promoter Calculator to calculate the five most predominant mRNA isoforms, ranked according to their predicted transcription initiation rates. For each distinct mRNA isoform sequence, we then separately calculated their structural and functional features (Fig. 3A). Altogether, we used these isoform-specific features to develop and test a machine learning regressor that quantitatively predicts steady-state mRNA levels and kinetic mRNA decay rates.

We began the model training and testing procedure by combining the isoform-specific features and mRNA level measurements (steady-state levels and 4 kinetic levels per mRNA) into an augmented dataset, following by randomized splitting into a training dataset (36,219 mRNAs) and an unseen test dataset (9064 mRNAs). Importantly, to develop a more generalized model, we randomly split the mRNAs in each design group into training and test datasets to ensure that the same fraction of each design group was uniformly represented in both datasets. Using the training dataset, we then carried out machine learning using a gradient boosting algorithm (LightGBM v3.3.2), followed by testing on the unseen test dataset. We first developed five independently trained models, one to predict steady-state mRNA levels and four to predict mRNA levels at each post-rifampicin kinetic timepoint (Fig. 3B).

Gradient boosting uses a learned decision tree with automatic feature categorization to develop an ensemble of weak models that are summed together to predict the outcome. LightGBM is an enhanced gradient boosting algorithm that rapidly sub-divides numerical features (e.g., the predicted translation initiation rates) into optimal discrete bins for improved model accuracy. LightGBM also quantities the importances of each isoform-specific feature, enabling the removal of features that do not affect the outcome. Through iterative development, we identified a single set of hyperparameters (Supplementary Table 2) and a single minimal set of 496 features (Supplementary Data 2) that yielded accurate models for both steady-state and kinetic mRNA levels, using log-transformed mRNA levels as the outcome (Fig. 3B). Model accuracy was highest when predicting steady-state mRNA levels (train $R^2 = 0.75$, test $R^2 = 0.69$) and mRNA levels after 2 min of rifampicin treatment (train $R^2 = 0.72$, test $R^2 = 0.65$), followed by reductions in accuracy as mRNA decay further reduced mRNA levels. The lowest model accuracy was observed at the longest kinetic timepoint (16 min after rifampicin treatment, $R^2 = 0.52$, test $R^2 = 0.43$). We then developed a machine learning model to predict a mRNA's decay rate by combining the isoform-specific features with the models' log-transformed mRNA level predictions. Using the same procedure and hyperparameters, we developed a LightGBM model with moderate accuracy on both the training and unseen test datasets (train $R^2 = 0.56$, test $R^2 = 0.43$) with absolute-time mRNA decay rate predictions that spanned a 10.3-fold range (half-lives from 0.7 to 9.8 min) (Fig. 3C). Overall, the model predicted mRNA decay rates with a median relative error of 18.5% when evaluated on the unseen test dataset.

During model development, we eliminated several candidate features that had low importances when predicting mRNA levels,

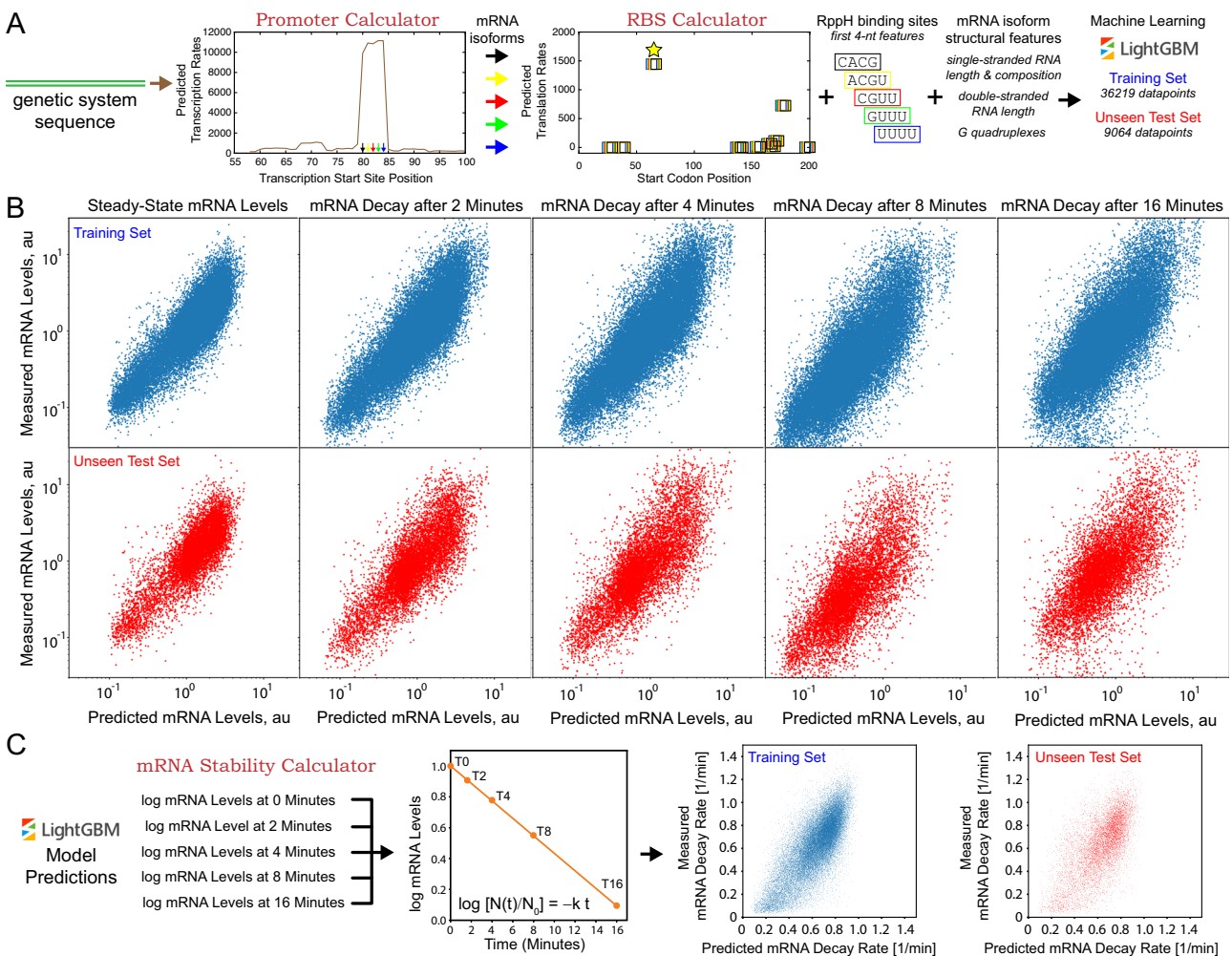

**Fig. 3 | A hybrid biophysical-machine learning model of mRNA decay. A** The Promoter Calculator and RBS Calculator biophysical models are used to predict the transcription initiation rates and translation initiation rates of the five most predominant mRNA isoforms, followed by calculating the isoforms' structural features. The biophysical features, measured mRNA levels, and measured mRNA decay rates of 45,283 characterized mRNAs are split into a training dataset ($N$ = 36,219) and an unseen test dataset ($N$ = 9064), followed by training and testing of a machine learning model (LightGBM). **B** Individually trained models accurately predicted mRNA levels at the steady-state timepoint (train $R^2$ = 0.75, test $R^2$ = 0.69) and post-rifampicin treatment timepoints (+2 min, train $R^2$ = 0.72, test $R^2$ = 0.65; +4 min, train $R^2$ = 0.68, test $R^2$ = 0.60; +8 min, train $R^2$ = 0.61, test $R^2$ = 0.53; +16 min, train $R^2$ = 0.52, test $R^2$ = 0.43). **C** The mRNA Stability Calculator accurately predicted mRNA decay rates by combining biophysical features and model-predicted mRNA levels at each timepoint (train $R^2$ = 0.56, test $R^2$ = 0.43).

including mRNA hairpin heights, the lengths of mRNA internal loops, and the number of mRNA bulges. We confirmed that changing these mRNA structural factors, while keeping other factors constant, did not significantly alter the mRNAs' measured decay rates (Supplementary Fig. 4). In comparison, the same calculations identified that the RppH binding sites, the CDS translation rates, the promoter transcription rates, and the ssRNA compositions were the most important features controlling the model's predictions in agreement with our multi-factor analysis (Supplementary Fig. 5). We also found that tracking only the single most predominant mRNA isoform, ranked according to their transcription rates, resulted in a statistically significant reduction in model accuracy (train $R^2$ = 0.72, test $R^2$ = 0.67; $p$-value of comparison =$1.4 \times 10^{-7}$). Finally, we verified that the model-predicted steady-state mRNA levels were similarly accurate across all design groups (Supplementary Fig. 6), suggesting that the model has high generalizability even when mRNAs have combinations of diverse sequence, structural, and functional features.

## Design rules for controlling mRNA decay rate

As we have shown, several interactions work together in a coupled fashion to collectively control mRNA decay rates, making it a challenge to directly analyze data and disentangle cause-effect relationships to use as design rules. However, by using the developed LightGBM model as a predictor, we can now carry out a systematic variation of the interactions and quantitatively determine how they each individually alter mRNA stability. We first selected a baseline mRNA with sequence, structural, and functional features that confer high mRNA stability, followed by systematically changing single feature values across the physiologically relevant range (Fig. 4). In this analysis, all five predominant mRNA isoforms are treated as having identical features to more clearly show cause-effect relationships. For example, we systematically varied the RppH binding site of the baseline mRNA, ranked them according to the predicted effect on the mRNA's steady-state level, and found that RppH binding sites with homopolymeric compositions (skewed statistics towards one nucleotide) greatly lowered mRNA stabilities as compared to RppH binding sites with balanced compositions (Fig. 4A). Accordingly, for a first design rule, simply changing the first four nucleotides of the mRNA transcript can be used to tune mRNA decay rates with small increments across a fourfold range (Supplementary Fig. 7).

We then systematically varied the baseline mRNA's translation initiation rate and found that the LightGBM model predicted a

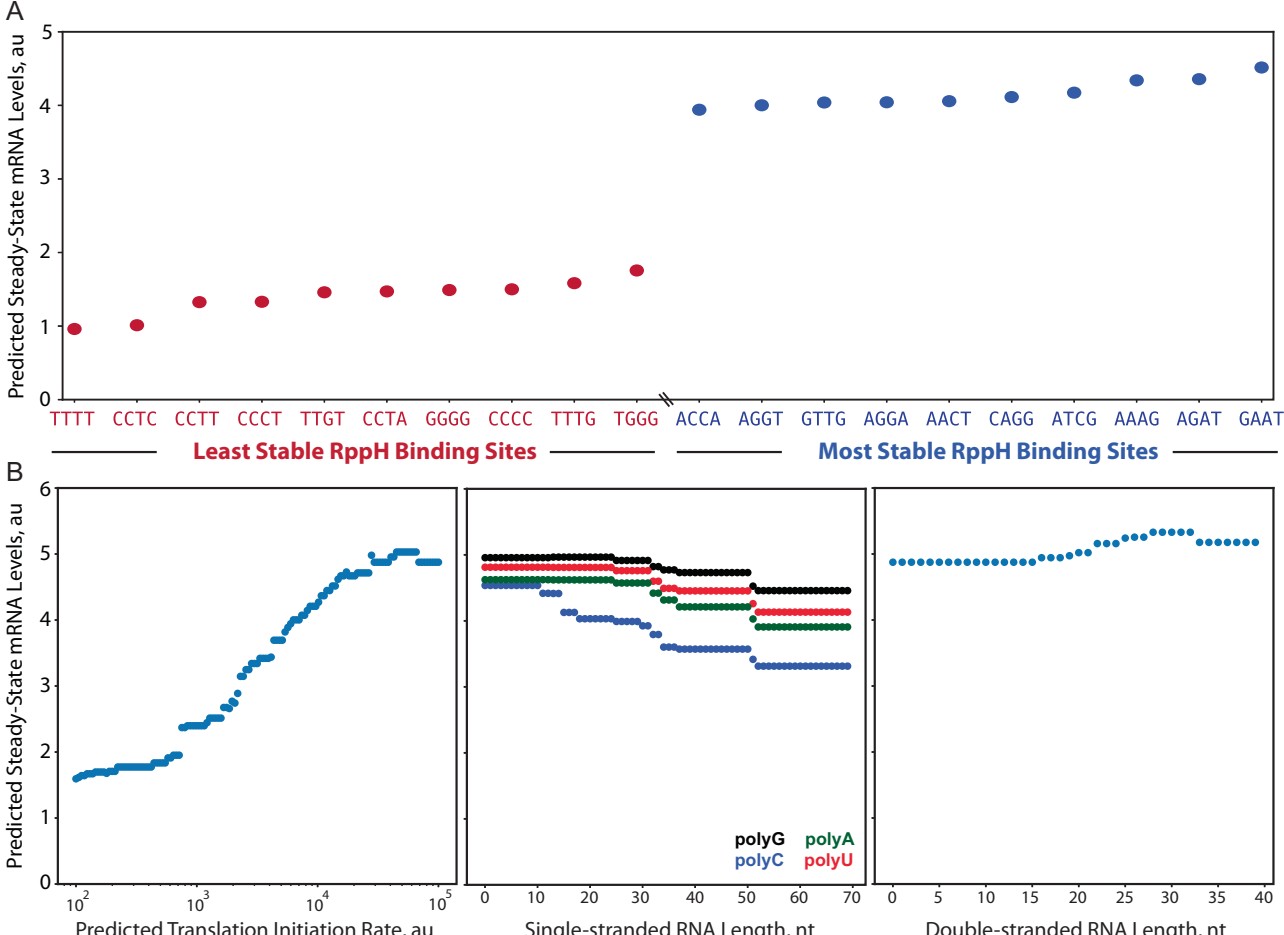

**Fig. 4 | Model-predicted design rules for controlling mRNA levels and decay rate. A** The RppH binding sites sequences ranked according to their model-predicted changes in steady-state mRNA levels. The top and bottom 10 RppH binding sites are shown that (red) decrease or (blue) increase mRNA stability.

**B** Model-predicted design rules for changing steady-state mRNA levels when varying (left) translation initiation rate of a CDS, (middle) the amount of single-stranded RNA in the 5′ UTR with either polyA, polyG, polyC, or polyU composition, or (right) the amount of double-stranded RNA in the 5′ UTR.

sigmoidal effect on the steady-state mRNA level with a ceiling plateau at around 50,000 au on the RBS Calculator 2.1 scale (Fig. 4B). Therefore, all other factors being equal, a higher translation rate will also stabilize a mRNA transcript, yielding higher protein expression levels, up to a ceiling plateau. The model also quantitatively shows that longer ssRNA regions in the 5′ UTR lead to lower mRNA stability, although the effect is most pronounced for C-rich and A-rich ssRNA sequences (Fig. 4B, middle). PolyA nucleic acids are known for being particularly straight and stiff, which can increase their accessibility to RNase activity[56]. Finally, we found only modest increases in mRNA stability when 5′ UTR regions contained more mRNA secondary structures, independent of other factors (Fig. 4B, right). Therefore, it's likely that mRNA secondary structures on their own do not protect mRNAs and, instead, it is the presence or absence of unstructured RNA that is the dominant design factor.

## Discussion

We designed 62,120 5′ UTRs to quantitatively determine how a mRNA's sequence controls its decay rate via combinations of sequence, structural, and functional features (Fig. 1A, B). Leveraging oligopool synthesis and library-based cloning, we introduced these 5′ UTRs into a nominal expression system, creating a barcoded cell library with very high coverage. We then measured both the steady-state mRNA levels and kinetic mRNA decay rates across the cell library, using rifampicin treatment to halt transcriptional initiation and deep sequencing to

quantitatively record mRNA levels across kinetic timepoints (Fig. 1C). Our 5′ UTR designs varied in vivo mRNA decay rates across the physiological range with mRNA half-lives varying from 20 s to 20 min (Fig. 1D), demonstrating the importance of the features used to design the 5′ UTRs.

Cloning and characterization of the cell library yielded exceptional coverage in terms of both breadth and depth (Supplementary Data 2). Over 99.8% of UTR variants were present in the cell library (UTR variants with at least one DNA-Seq read after deep sequencing), though we opted to stringently consider only UTR variants that had at least 100 T0 reads to improve measurement precision, yielding a breadth coverage of 91.5%. The depth coverage was also exceptionally high with median read counts of 1034 to 2271 across the timepoints. The higher read counts enabled us to measure mRNA decay rates with more quantitative precision. Over 80% of the characterized UTR variants exhibited purely exponential mRNA decay, which is expected when there is only one rate-limiting step to triggering processive mRNA degradation.

Notably, when we quantified mRNA decay rates, it was essential to apply the RNA spike-in controls for normalization of the kinetic mRNA levels. When carrying out typical RNA-Seq measurements, the total RNA mass across a cell library is about the same across samples even if individual variants have higher or lower relative frequencies. However, after subjecting our cell library to rifampicin treatment, we expected and found that the total RNA mass was greatly reduced, particularly at

the 16-min timepoint, invalidating that typical assumption (Supplementary Data 1). We accordingly used the RNA spike-in control read counts to normalize the frequencies of the designed UTR variants across timepoints; the RNA spike-in control read counts were about tenfold higher at the 16-min timepoint as compared to the initial timepoint.

As another notable finding, we found that differences in a UTR variant's steady-state mRNA level were not indicative of its mRNA decay rate. Typically, gene expression models assume that the steady-state mRNA concentrations depend on the ratio between transcription initiation rates and mRNA decay rates such that there should be a proportional relationship between a mRNA's decay rate and the inverse of its steady-state mRNA level. Here, we found an unexpected non-linear, sigmoidal-like relationship between the UTR variants' inverse steady-state mRNA levels and mRNA decay rates (Supplementary Fig. 8). The deviation from the expected proportional relationship could arise in at least two ways. First, differences in the promoters' transcription initiation rates will affect the steady-state mRNA levels but do not in principle alter the mRNA's exponential decay kinetics, which adds a confounding factor to the comparative analysis. Second, for some UTR variants, it is possible that high mRNA concentrations could saturate the RNA degradosome machinery, converting what is normally a first-order decay process into a zero-order decay process. About 20% of the UTR variants exhibited kinetic mRNA levels with large deviations from a first-order decay process (Supplementary Data 1). Regardless, these findings motivate the collection of mRNA decay measurements rather than relying solely on measuring steady-state mRNA levels.

Overall, a key conclusion of this work is that the sequence determinants that control mRNA decay are highly entangled; small changes to the 5′ UTR sequence can alter both structural and functional features, affecting the strengths of multiple interactions at the same time, making it challenging to directly elucidate cause-effect relationships. For example, a few nucleotide changes can alter both a mRNA's structure as well as its translation rate, both affecting mRNA decay rates via distinct and potentially opposing mechanisms. Multi-factor categorization could somewhat disentangle individual contributions (Fig. 2), suggesting that cause-effect relationships can be learned by training an ensemble of models and combining their predictions together. Here, we combined biophysical models with machine learning to develop a predictive model of mRNA decay (Fig. 3). The LightGBM algorithm used the biophysical model predictions to disentangle the mRNA sequence into its structural and functional features, enabling the algorithm to learn how these interactions control mRNA decay rates. We then systematically queried the model to extract these cause-effect relationships, revealing the design rules that control mRNA decay rates (Fig. 4). Notably, a LightGBM model trained using only sequence information was less accurate (steady-state: train $R^2 = 0.65$, test $R^2 = 0.61$; decay rate: train $R^2 = 0.43$, test $R^2 = 0.37$), while not yielding any mechanistic design rules.

We found four types of interactions that primarily controlled mRNA stability and decay rates. The first four nucleotides of a mRNA can alter mRNA stability by up to fourfold (Fig. 4A). RppH binds to the 5′ end of a mRNA transcript and converts the triphosphate 5′ end into a monophosphate in a multi-step reaction, accelerating end-dependent mRNA decay[40]. Changes to the first four nucleotides modulate RppH's binding affinity[39,57,58]. Independent of mRNA secondary structure, the RppH binding sites with the fastest mRNA decay rates contain homopolymeric regions, suggesting that stacking interactions between end-terminal nucleotides are increasing RNA stiffness and accessibility, which could improve RppH binding affinities.

Second, the mRNA's translation initiation rate can alter its stability by over 10-fold (Fig. 2A) with a sigmoidal cause-effect relationship (Fig. 4B). This purely data-driven relationship is highly consistent with the predictions of a previously developed biophysical model of

ribosome protection that directly relates a mRNA's predicted translation initiation rate to its mRNA decay rate[4]. In this biophysical model, we used statistical physics to calculate how changing the translation initiation rate affects the average distance between elongating ribosomes, which controls accessibility to RNase activity. It is notable that both efforts arrived at similar sigmoidal relationships even though they applied different measurement techniques on different sets of mRNAs. However, in both efforts, only the translation initiation rates were varied, while the translation elongation rates were kept relatively constant. According to the biophysical model, we expect that the true design factor is the ratio between the translation initiation rate and the translation elongation rate, which are nonetheless being systematically varied in both efforts.

Third, longer regions of unstructured mRNA in the 5′ UTR lowered mRNA stability by creating more landing pads for RNase E/G binding and more potential cleavage sites for initiating processive mRNA degradation (Fig. 4B). Homopolymeric C-rich and A-rich ssRNA regions lowered mRNA stability more so than U-rich and G-rich sequences, suggesting that sequence composition has a direct impact on the persistence length of ssRNA and its accessibility to RNase binding.

Finally, through our analysis and model development, we found that mRNA structures only provided protection from RNase activity by lowering the lengths of ssRNA regions. When keeping the amount of unstructured mRNA constant, the addition of mRNA duplexes and hairpins did not improve mRNA stability (Fig. 4B). The introduction of G-quadruplex RNA structures, but not i-motif RNA structures, increased mRNA stability by reducing the amount of unstructured RNA and blocking RNase binding (Fig. 3F, G). Notably, the sequence motif $[3G-6N]_X$ is highly degenerate and can be used to form protective G-quadruplexes when the formation of other mRNA structures is not possible.

The effects of most of these interactions are likely to be conserved across many bacterial species as they depend mainly on the biophysics of RNA folding and foundational binding mechanics. The enzymes in the RNA degradosome are well-conserved across most bacterial species; for example, *E. coli* RNase E and RppH are highly structurally similar to their homologs across the Pseudomonadota phylum, according to the AlphaFold Cluster Database[59]. As we found that *E. coli* RNase E does not seem to have a sequence-specific binding motif, but instead binds to unstructured single-stranded mRNA regions, it is highly likely that this interaction and its effect on mRNA decay are similar across many bacterial species. Likewise, increasing the translation rate of a mRNA (up to a point) is expected to increase its mRNA stability across most bacterial species as the protective effects of bound ribosomes are well-conserved. However, *E. coli* RppH makes sequence-specific contacts with the 5′ terminal end of mRNA and thus its sequence specificity is less likely to be conserved across phylogenetically dissimilar bacterial species.

We developed the predictive model of mRNA stability—named the mRNA Stability Calculator—to quantitatively fill key gaps in the sequence-function relationships that control gene expression levels. Notably, the same sequence determinant can affect multiple gene expression steps. For example, the first four nucleotides of a mRNA transcript will affect both its transcription initiation rate and its decay rate. A mRNA's UTR region controls both its translation initiation rate and its decay rate. If not considered, these entanglements will have a misleading effect on analysis outcomes. Accordingly, we applied predictive models of transcription and translation to disentangle these sequence determinants and to develop a comprehensive of mRNA decay. By combining these models together, we now have the potential to accurately predict sequence-function relationships across diverse bacterial genetic systems. The developed models and design rules have been combined with sequence optimization algorithms to automatically design genetic systems with targeted transcription, translation, and mRNA decay rates. The source code for training and

testing the mRNA Stability Calculator is available at https://github.com/hsalis/SalisLabCode. A web interface to our integrated design platform for engineering organisms is available at https://salislab.net/software.

## Methods

### Library design and cloning

62,120 5′ UTR sequences were designed to systematically vary sequence, structural, and functional determinants. Each UTR variant was assigned a unique 15-nucleotide barcode sequence with pairwise Hamming distances greater than 2. 62,120 oligonucleotides were then designed, using the same pair of designed primer binding sites and four restriction sites to facilitate PCR amplification of the oligopool and two-step library-based cloning into the vector. Restriction sites for AvrII and AatII are used to insert the oligopool into the vector to create an intermediate plasmid library. Restriction sites SacI and EcoRI are used to insert a sfGFP CDS into the intermediate plasmid library to create the final plasmid library. Padding sequences were added to ensure that all oligonucleotides are 170-nucleotides long. The oligopool was split into five equal portions and synthesized by Genscript. All designed UTR sequences and constant regions are listed in Supplementary Data 1.

The five oligopools were separately amplified by PCR using 22 cycles, followed by gel separation, and extraction. The five fragment mixtures and a pFTV1-derived vector were separately digested using AvrII and AatII (37 °C for 6 h), purified, ligated (300 fmoles digested oligopool per 30 fmoles digested vector), and transformed into *E. coli* DH5α cells, yielding at least one million CFUs. The transformed cell library was cultured in selective LB media and the intermediate plasmid library was purified. The intermediate plasmid library and a PCR-amplified sfGFP CDS cassette were then digested using EcoRI and SacI (37 °C for 6 h) and purified. The digested intermediate plasmid library was treated with shrimp alkaline phosphatase to prevent self-ligation, followed by ligation to the sfGFP cassette (90 fmoles sfGFP, 25 fmoles intermediate plasmid library) and transformation into *E. coli* DH5α cells, yielding at least 10 million CFUs. Cell libraries from each of the five oligopools were equally mixed according to their CFUs and then cultured in selective LB media, followed by cryostocking one portion and carrying out plasmid purification on another portion. Cell library coverage was determined by MiSeq next-generation sequencing on the PCR-amplified barcode region of the plasmid library. After construction, the plasmid library is composed of a vector (ColE1 origin, a chloramphenicol resistance marker), a constant upstream J23100 promoter, a variable 5′ UTR region, a constant sfGFP CDS, a variable barcode region, a constant RBS and CDS that expresses mRFP1 in the second position of the bacterial operon, and a constant double transcriptional terminator. The downstream RBS was designed to contain an insulating hairpin with a predicted translation initiation rate of 9973 au (RBS Calculator v2.1) expressing mRFP1 at a moderate level.

### Massively parallel mRNA level measurements

An entire cryostock was resuspended in 100 mL of LB media supplemented with 50 µg/mL Chloramphenicol (Cm) in a 1 L Erlenmeyer flask and incubated at 37 °C, 300 RPM agitation for 12 h. Afterwards, it was diluted to an $OD_{600}$ of 0.05 in 50 mL EZRich media (Teknova) supplemented with 50 µg/mL Chloramphenicol (Cm) in a 1 L Erlenmeyer flask, and incubated at 37 °C, 300 RPM agitation for 2 h until the $OD_{600}$ reached 0.3. Again, the culture was diluted to an OD600 of 0.05 in 250 mL EZRich media supplemented with 50 µg/mL Chloramphenicol (Cm) in a 1 L Erlenmeyer flask and incubated at 37 °C, 300 RPM agitation for 1.5 h until the $OD_{600}$ reached 0.25 and the cells were harvested. All characterization steps were done in triplicate. Three 50 mL aliquots were taken and used to perform a plasmid extraction using the plasmid DNA mini kit (Omega Biotek). Three

4 mL samples of the culture were taken and mixed with 8 mL of RNAprotect (Qiagen) to fix cells and stop RNA degradation. These samples were processed to determine the cell library composition and mRNA levels at the T0 timepoint. Rifampicin was then added to the remaining bacterial culture to a final concentration of 500 ng/µL to halt transcriptional initiation. Three 4 mL samples were each taken at 2, 4, 8, and 16 min after the addition of rifampicin, followed by mixing with 8 mL RNAprotect. Fixed cultures were then pelleted at $5000 \times g$ for 10 min, followed by RNA extraction (Total RNA Purification, Norgen Biotek) and removal of contaminant DNA (Turbo DNase, Ambion).

Spike-in control RNA was produced using HiScribe T7 High Yield RNA Synthesis (NEB) from a linear DNA template that uses a T7 promoter to produce mRNA of similar size to the designed mRNAs with identical sites for cDNA synthesis and PCR amplification of the barcode region, though using a distinct barcode sequence. The concentrations of the extracted total RNA from the cell library and the spike-in control RNA were measured with the Quant-iT RNA assay (Thermo Fisher). Spike-in control RNA and total RNA were combined at the ratio of 10 attomoles of spike-in control to 1 µg of total RNA. Samples were then depleted of rRNA (NEBNext rRNA depletion, NEB), followed by cDNA synthesis (SuperScript IV first-strand synthesis, Invitrogen) using a reverse transcript-specific primer that binds upstream of the barcode region. Barcoded amplicon library generation for DNA-Seq and RNA-Seq were then carried out via PCR amplification of the barcode region of the either the purified plasmid DNA or the purified cDNA, respectively, using 25 cycles and gel extraction. DNA amplicon libraries were then submitted for deep sequencing, which includes adapter library preparation and flow cell loading (Illumina NovaSeq, Genewiz). Read counts were determined from FASTQ files by mapping and counting barcode sequences, employing a hash look-up table and multi-threaded operation. All barcode sequences were designed to have pairwise Hamming distances greater than 2, enabling unique barcode mapping with at most one nucleotide mutation.

### Data analysis and model development

mRNA decay rates were determined from the DNA-Seq read counts, RNA-Seq read counts, and spike-in control RNA read counts at the 0, 2, 4, 8, and 16 min post rifampicin timepoints. RNA-Seq read count ratios ($R_i$) were calculated according to $R_i = N_i / N_O / CR_i$, where $N_i$ are the read counts at each timepoint, $N_O$ are the read counts at the 0 timepoint, and $CR_i$ are the spike-in read count ratios at each timepoint. $CR_i$ is calculated using $CR_i = C_i / C_O$, where $C_i$ are the spike-in read counts at each timepoint and $C_O$ is the spike-in read count at the 0 timepoint. We then fit the time series $R_O$, $R_2$, $R_4$, $R_8$, and $R_{16}$ to an exponential decay curve with a single unknown parameter ($k$). We use non-linear least squares regression to identify the best-fit value of $k$, using the SciPy curve_fit function. We quantify the goodness of fit using Pearson $R^2$. The source code for this analysis is found in Supplementary Information.

LightGBM models to predict mRNA levels at each timepoint were independently trained and tested using the natural log of the $RNA_i / DNA_i$ ratio at the outcome, where $RNA_i$ are the RNA-Seq read counts at each timepoint and $DNA_i$ are the DNA-Seq read counts at each timepoint. UTR variants from each design group were uniformly and randomly sampled to create the 80%/20% train-test split. LightGBM were trained using hyperparameters listed in Supplementary Table 2. All sequences, features, outcomes, and model predictions for the train and test datasets are included in Supplementary Data 2. Using the same hyperparameters and train-test datasets, the LightGBM model to predict mRNA decay rates was trained and tested using the mRNA decay rate (1/min) as the outcome and an augmented feature set that included the natural log of mRNA level predictions at the 0, 2, 4, 8, and 16 timepoints.

**Reporting summary**

Further information on research design is available in the Nature Portfolio Reporting Summary linked to this article.

## Data availability

All sequences and experimental measurements are available in Supplementary Data 1. All featurization, model calculations, and model predictions are available in Supplementary Data 2. Next-generation sequencing read data files in fastQ format are publicly available at NCBI with accession identifier PRJNA1177892.

## Code availability

A Python source code implementation of the model and all data needed to reproduce model development are available at https://github.com/hsalis/SalisLabCode.

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

## Acknowledgements

This project was supported by funds from the Air Force Office of Scientific Research (FA9550-14-1-0089), the Defense Advanced Research Projects Agency (FA8750-17-C-0254), and the Department of Energy (DE-SC0019090) to HMS.

## Author contributions

D.P.C., A.H., and H.M.S designed the study and analyzed results. D.P.C. and H.M.S. wrote the manuscript. D.P.C. and G.E.V. conducted the experiments.

## Competing interests

H.M.S. is a founder of De Novo DNA. D.P.C., A.H., and G.E.V declare no competing interests.
