## [Transparent Peer Review file · Nature Communications]

Predicting synthetic mRNA stability using massively parallel kinetic measurements, biophysical modeling, and machine learning

Corresponding Author: Professor Howard Salis

Version 0:

Reviewer comments:

Reviewer #1

(Remarks to the Author)

The authors present an interesting study of bacterial mRNA degradation kinetics and stability with the aim of predicting the mRNA degradation rate from its sequence. They used a large library-based approach to perform massively parallel kinetic measurements for >60,000 mRNA designs with a variety of 5' UTR features and sequences. They created a machine learning model that also uses their previously published RBS Calculator and Promoter Calculator biophysical models to predict the mRNA decay rate constant and steady-state mRNA level. They report some important features of the mRNA 5' UTR that have effects on the mRNA degradation rate, which is quite interesting. I appreciate these aspects of the work and agree that the ability to accurately predict the degradation rate of an mRNA transcript from its sequence would be a great advancement for field and for predictive design and modeling of various genetic systems.

Below I have some comments to improve the manuscript.

The presentation of the model as a platform computational tool seems to suggest that it is expected to be accurate for other bacteria and generically for a given mRNA sequence. However, it may only be intended to be used for E. coli DH5-alpha. What bacteria would you expect this model to apply to? Is the model limited to synthetic mRNAs, which is suggested from the title? What are requirements for the mRNA for the model as presented to be applicable? If there are anticipated limitations or requirements for the model, it would be appropriate to include them in the Discussion.

The results of the MiSeq library sequencing suggest that only 59,721 mRNAs were constructed, rather than the 62,120 reported in the abstract and introduction. If this is correct, it seems more accurate to revise the language.

The barcode (15 nt) was placed downstream of the CDS to be outside of the UTR, and only the barcode amplicon was sequenced to quantify the DNA and RNA concentration. This assumes that there is a perfect pairing between the barcode and UTR sequence. Was any type of sequencing performed to determine the percentage of constructs that had an identical match for the full UTR sequence and the barcode for the constructed library?

The methods for the next-generation DNA and RNA sequencing and analysis are too brief to be reproduced by others. As described, "Amplicon libraries were then characterized using deep sequencing (Illumina NovaSeq, Genewiz)." I assume that the subsequent data analysis uses only the perfect match reads as the "mapped" reads. However, it is not stated in the methods.

It is not clear from the Methods if all kinetic measurements were performed in E. coli DH5-alpha. If that is correct, it would be good to briefly justify choosing this strain and discuss what effects it may have on using the model for other E. coli strains.

Were the kinetic measurement assays performed more than once to test reproducibility of the reported data and reported mRNA degradation kinetic constant for each UTR design? It seems that the model would be less accurate if the dataset is a single experiment.

For the reported conclusions relating mRNA features to degradation rate (Figure 2B–2G), the statistical analyses to justify the conclusions seem to be missing.

Another related point for the statistical analysis is that there is not a description or section in the Methods for the statistical analyses that were performed.

The model testing datasets were subsets (20%) of the designed mRNA library with a random split of the mRNAs in each design group. However, the set of mRNA designs contained many highly similar UTR sequences. I would expect that the model would have accurate predictions if it were trained by highly similar sequences. Were the models used to predict the mRNA degradation rate of other transcripts that differ more greatly, such as those from other papers containing different promoters or CDSs, as a more generic test?

It was concluded that using only the single most predominant mRNA isoform lowered the model's accuracy of the steady-state mRNA level. However, if I am interpreting correctly, the supporting data analysis is a decrease of the testing dataset R-squared from 0.69 to 0.67. It should be discussed whether this is a significant change. I assume that if you randomly split the data multiple times, there would be some variation of these values.

The reported non-linear relationship between the mRNA degradation rate and the inverse of the steady state mRNA concentration (reported in the Discussion and Supplementary Figure 8) is interesting, yet this conclusion is not supported by statistical analysis.

The Methods do not seem to mention how the measured steady-state mRNA levels used for the model training and testing in Figure 3B were determined and whether they are directly from the experimental data. The reported units for mRNA concentration are arbitrary units (au), though it is not clear how these quantities were determined.

In line 475, the phrase “highly polymeric” is used to describe the subset of mRNAs. I am not familiar with what this means, and perhaps instead you mean that they contain homopolymeric regions or tracts. This also comes up in line 493.

I did not find the mRNA Stability Calculator web interface available at the website provided. I assume you may be waiting until publication to release it.

The Introduction would be improved by expanding the discussion of bacterial mRNA degradation models that have been developed by other researchers.

Reviewer #2

(Remarks to the Author)
Review of Cetnar et al

This study carries out a systematic analysis of mRNA stability in *E. coli* by coupling high-throughput data collection with machine learning. In brief, the authors used massively parallel sequencing to measure mRNA-specific degradation rates for a large library of carefully designed, sequence-diverse mRNAs (62,120), employed rational multi-factor analyses and biophysics-guided machine learning methods (i.e., a gradient-boosting framework) to examine the molecular determinants of mRNA stability within their library, and developed predictive algorithms to estimate time-dependent mRNA levels (i.e., mRNA levels at specific time points post inhibitor-mediated transcriptional arrest) and degradation kinetics. Overall, the large mRNA library, which includes half-lives that vary by 10- to 100-fold, was sufficiently variable to permit an assessment of the isolated influence—or lack thereof—of important design factors (e.g., translation rates, secondary structures, sequence compositions, G-quadruplexes, i-motifs, and RppH activity); it also enabled the development of models with reasonable predictive power. The paper is well-written, easy to follow, and may be appropriate for *Nature Communications*. In its present form, however, the study makes observations and conclusions that are lacking in statistical significance (i.e., they are based on plots where differences between data points are not clear and/or lack a statistical test), and the generalizability of the models across genetic contexts and growth conditions is not assessed. These weaknesses can probably be remedied.

Major Points

1. Line 97: protein coding sequence (CDS). The entire paper is based on one CDS (GFP), which is understandable. The CDS, however, can affect mRNA stability—at least in human cells (see reference). Of course, repeating the entire analysis with a different protein is beyond the scope of this study, but because the study already identifies a range of mRNA stabilities, it would be informative to compare a subset of mRNAs (e.g., 10) with a few different proteins (e.g., GFP, maltose binding proteins, perhaps an inactive enzyme). Are the mRNA half-lives the same, regardless of CDS?

Ref: *Nature Communications* volume 13, Article number: 6829 (2022)

2. Related to the above. It seems logical that growth conditions, specifically those that might elicit some form of cellular stress, could re-weight the relative contributions of different mRNA attributes (e.g., RppH activity). It would be informative to take a subset of mRNAs with different half-lives (e.g., 10) and, ideally, different half-life-affecting features, and evaluate them in a new growth condition. Do the half-lives change between growth conditions? Do they correlate between conditions? This analysis, like the one above, would go a long way in establishing generalizability. Importantly, even if the results are not generalizable—across CDSs or growth conditions—the findings would be valuable and increase the impact of the study.

3. Line 99: complete paragraph: I found this paragraph and the following paragraph to be somewhat confusing. What is being held constant and what is being changed within each design group? Could the authors clarify further? The wording is a little confusing.
4. Line 231: "we find a clear inverse relationship between the CDSs' predicted translation initiation rates and their measured decay rates (Pearson R = -0.31)." This trend does not look significant. At most, it is a very weak correlation. Could the authors comment on the significance of the trends?
5. Line 245: "PolyA motifs appeared to have a larger impact on mRNA decay than other polymeric motifs, particularly when mRNAs had high predicted translation rates (Figure 2E)." How so? The mRNA decay rates for mRNAs with polyA motifs were highly variable (more so than for polyG). Wouldn't that imply that polyA tails do not strongly impact decay rates?
6. Line 248: "Finally, we found that G-quadruplex tertiary structures had a strong protective effect on mRNA stability, lowering their decay rates, while the i-motif tertiary structures had not apparent protective effect on mRNA stability (Figure 2FG)." This statement is not supported by the data. In Figs. 2F-2G the differences between the blue and orange points do not appear to be statistically significant. Could the authors justify their statement with a statistical test. Perhaps, they could find a window where it is true (e.g., ssRNA length of 30-40 for G-quadruplex).
7. Lines 260-263: same comment as above. Please check the statistical significance of the observations made about Figs. 2F-2G.
8. Line 383: "found that RppH binding sites with polymeric compositions (skewed statistics toward one nucleotide) greatly lowered mRNA stabilities as compared to RppH binding sites with balanced compositions (Figure 4A)." Based on the data, could it be equally likely that A-rich binding sites improve stability? The least stable binding sites are A poor (i.e., there are no polymeric A binding sites in this set).
9. Line 394: "The model also quantitatively shows that longer single-stranded RNA regions in the 5' UTR lead to lower mRNA stability, although the effect is most pronounced for C-rich and A-rich ssRNA sequences (Figure 3B)." Both statements seem to lack statistical significance. The impact of the single-stranded region seems extremely modest (i.e., nearly a flat line) and the differences between polyA and polyU or polyG seem insignificant. PolyC is a bit more influential, but is it significant? Can the authors use a statistical test to substantiate the importance of the trends that they are calling out? A reference that would justify difference in stability as impactful/not impactful, from a biological perspective, could also help.
10. Line 421: "Over 80% of the characterized UTR variants exhibited purely exponential mRNA decay." What test are the authors using to compare models? Did they compare different kinetic equations for decay?
11. Line 491: The above point comes up again as one of the major conclusions of the paper. "Polymeric C-rich and A-rich single-stranded RNA regions lowered mRNA stability more so than U-rich or G-rich sequences, suggesting that sequence composition has a direct impact on the persistence length...." These differences do not seem significant.

Minor Points

1. Line 84: Fig. 1C. This plot lacks a legend. To what do the different colors correspond? How were the relevant samples chosen? Are there estimates of error (for points or fits)?
2. Line 153: How did the authors pick the dosing level / concentration of Rifampicin? I ask only because a high concentration could elevate cellular stress (which may matter).
3. Line 165-178: Much of this section seems like it could be moved to Methods.

Version 1:

Reviewer comments:

Reviewer #1

(Remarks to the Author)

The authors have improved the manuscript and adequately addressed the review comments. I believe the manuscript is now suitable for publication in Nature Communications and will be significant for the field.

Reviewer #2

(Remarks to the Author)

There is still a major disconnect between (i) the data and (ii) the words used to describe it. Unfortunately, statements about data / trends tend to be propagated through citations more easily than actual data; it is important for the text to be clear, accurate, and informative. The authors can remedy this quickly. Please see attached pdf.

Review of Cetnar et al

The reviewers have done a nice job responding to my questions; clarifying changes to the text, however, were minimal—a perplexing decision, given the thoughtful replies. The text would be clearer, and the analysis, more rigorous, if the authors incorporated elements from their responses into the actual paper. In general, there is a significant disconnect between the data and the words used to describe the data; the text still tends toward a blunt, overly broad interpretation of extremely weak

trends: There are several key opportunities for quick remedies:

1. Responses to prior comments 6 and 7, which relate to Figures 2F and 2G. The authors claim that they “ran statistical analyses on all relationships shown in Figure 2” and that “these calculations are now all reported in the manuscript text”, but they are not—not for 2F and 2G, where the relationships are suspect. See below.

Questionable statements from text:

- “Finally, we found that G-quadruplex tertiary structures had a strong protective effect on mRNA stability, lowering their decay rates, while i-motif tertiary structures had no apparent protective effect on mRNA stability (Figure 2FG).”
- “We found that the introduction of G-quadruplexes into the 5' UTR lowered the mRNAs' decay rates (to a constant of about 0.2 1/min) regardless of the secondary mRNA structures that would otherwise form with varied amounts of ssRNA (Figure 2F). In contrast, using the same analysis, we found that the introduction of i-motifs could greatly increase mRNA decay rates, compared to an equivalent mRNA that only formed secondary mRNA structures (Figure 2G).”

Comment: “Strong protective effect” is an absurd overstatement. Look at the plots. Again, the authors mention a statistical test, but they do not actually include that test in the figure caption. Also, it is not clear how they are calculating the two-tailed T Test, which is mentioned only in their response, as it relates to this data, not in the actual test. It is OK to say “weak, but statistically significant protective effect for intermediate lengths”. Let the data speak.

2. Responses to comments 4 and 5.

As above, the main text statements still overstate the trends. See below:

Questionable statements from text:

- “We find a clear inverse relationship between the CDSs' predicted translation initiation rates and their measured decay rates (Pearson $R = -0.31$, $N = 1527$ mRNAs) with higher translation rates providing mRNAs with more stability (Figure 2A), confirming that some design factors may only have a clear effect on mRNA decay rates when other contributing factors are kept relatively constant.”
- PolyA motifs appeared to have a larger impact on mRNA decay than other homopolymer motifs, particularly when mRNAs had high predicted translation rates (Figure 2E).

Comment: Look at the data—it doesn't match the description. The trends are still extremely weak. It would be clearer—and, notably, more informative—to say this: “We see a very weak, inverse relationship between...”. For 2E, it seems like the variability in the effect of the PolyA motifs is the more interesting result. The mean decay rates may be higher, but it does not look statistically significant, and the statistical tests, again, are not present in the relevant caption E.

Predicting synthetic mRNA stability using massively parallel kinetic measurements, biophysical modeling, and machine learning

Daniel P. Cetnar¹, Ayaan Hossain², Grace E. Vezeau³, and Howard M. Salis^{1,3,4*}

NCOMMS-24-01006

We would like to express our deep appreciation for the reviewers' constructive comments, which pointed out several areas where the manuscript could be improved. Below, in our point-by-point response, we answer the reviewers' questions and describe the improvements to the manuscript based on their feedback. The reviewers' questions are listed below, followed by our responses (**green**). We've also included a version of the manuscript with changes highlighted (in the peer review package).

Reviewer 1

The authors present an interesting study of bacterial mRNA degradation kinetics and stability with the aim of predicting the mRNA degradation rate from its sequence. They used a large library-based approach to perform massively parallel kinetic measurements for >60,000 mRNA designs with a variety of 5' UTR features and sequences. They created a machine learning model that also uses their previously published RBS Calculator and Promoter Calculator biophysical models to predict the mRNA decay rate constant and steady-state mRNA level. They report some important features of the mRNA 5' UTR that have effects on the mRNA degradation rate, which is quite interesting. I appreciate these aspects of the work and agree that the ability to accurately predict the degradation rate of an mRNA transcript from its sequence would be a great advancement for field and for predictive design and modeling of various genetic systems.

We thank the reviewer for their positive feedback. We agree that many interactions that affect mRNA degradation rate are often overlooked even though they can greatly affect the function of an engineered genetic system. The development of this model will fill a gap in our knowledge and provide an important building block in our overall plan to develop an integrated multi-model prediction & design platform for engineering genetic systems.

The presentation of the model as a platform computational tool seems to suggest that it is expected to be accurate for other bacteria and generically for a given mRNA sequence. However, it may only be intended to be used for E. coli DH5-alpha. What bacteria would you expect this model to apply to? Is the model limited to synthetic mRNAs, which is suggested from the title? What are requirements for the mRNA for the model as presented to be applicable? If there are anticipated limitations or requirements for the model, it would be appropriate to include them in the Discussion.

Yes, the "cross-species generalization" question is very important and though we often develop & validate biophysical models in one organism, our models are commonly used

in many others. In this work, we measured steady-state mRNA levels and kinetic mRNA decay rates for over 50000 synthetic (designed) mRNAs when expressed in *E. coli* DH5-alpha cells, grown in EZ Rich defined media during the exponential phase of growth. The objective here is to characterize the biophysical interactions that control how differences in mRNA sequence alter their degradation rates. Importantly, these biophysical interactions and their effect on mRNA decay are not unique to *E. coli* DH5-alpha, but are likely well-conserved across many bacterial species.

For example, we found that increasing a mRNA's translation initiation rate greatly decreases its decay rate, which happens when a mRNA is covered by more protective ribosomes and prevents RNases from cleaving it. We also found that a longer unstructured mRNA region increases the mRNA's decay rate, which increases the number of binding sites for RNases. These biophysical interactions will occur in many bacterial species as they rely solely on the thermodynamics of mRNA folding and fundamental translation processes, which are widely conserved. Other biophysical interactions, such as the RppH's binding affinity to the first four nucleotides of a mRNA transcript, are more likely to quantitatively vary across phylogenetically distant bacteria where a RppH homolog exists.

A key reason why we incorporate biophysical interactions directly into our model formulation is to make it possible to distinguish between interactions that are likely to be highly conserved versus other interactions that are likely to vary across evolutionary distant bacterial species. We expect that some model-predicted design rules will be "universal" across most bacterial species (**Figure 4B**) whereas others may only be 100% accurate at the phylum level (e.g. **Figure 4A**). Importantly, for distantly related bacteria, even an imperfect prediction is better than no prediction at all.

We've added a concise discussion of cross-species model generalization in the manuscript's Discussion section, summarizing these points.

The results of the MiSeq library sequencing suggest that only 59,721 mRNAs were constructed, rather than the 62,120 reported in the abstract and introduction. If this is correct, it seems more accurate to revise the language.

Yes, the initial MiSeq sequencing of the plasmid library yielded 7.9 million reads and 59721 expected barcode sequences with at least one read count. However, when sequencing the same plasmid library using NovaSeq sequencing, we received 1.9 billion reads (about 300 million reads per sample) and identified 61997 expected barcode sequences in the T0 DNA sample (only 123 missing) and 62104 expected barcode sequences in the T0 RNA sample (only 16 missing), again each barcode having at least one read count. These relatively small differences in library coverage are simply due to the depth of sequencing and random chance. These numbers are listed in the **Supplementary Data 1** file (Statistics tab).

However, from our perspective when developing & validating a model, it is more important to set a higher bar for measurement precision and only use datapoints with higher read

counts (>100) and exponential decay dynamics. From the NovaSeq deep sequencing, we found 56816 DNA barcodes and 58080 mRNA barcodes at the T0 timepoint that had at least 100 reads. Using the well-measured mRNA levels at the 5 timepoints, we found 50048 mRNAs that exhibited purely exponential decay dynamics with a single half-life. We then removed extreme (non-physiological) outliers and used 45283 datapoints to train and validate our model (36219 mRNAs in the training dataset and 9064 mRNAs in the unseen test dataset).

With all this being said, we modified the abstract to “Here, we carried out massively parallel kinetic decay measurements on **over 50000** bacterial mRNAs” (change bolded). In one perspective, we carried out measurements on all bacterial mRNAs produced by the plasmid library, but we only used the high-quality measurements to develop & validate the model. All measurements are in the Supplementary Data file.

The barcode (15 nt) was placed downstream of the CDS to be outside of the UTR, and only the barcode amplicon was sequenced to quantify the DNA and RNA concentration. This assumes that there is a perfect pairing between the barcode and UTR sequence. Was any type of sequencing performed to determine the percentage of constructs that had an identical match for the full UTR sequence and the barcode for the constructed library?

Yes, we had carried out preliminary sequencing of the first step of the 2-step cloning procedure via Sanger sequencing of miniprep plasmid from isogenic colonies, and found that 100% of the sampled barcodes were paired with their corresponding UTR sequence. This is fully expected as the non-random (designed) barcode sequences were physically part of the oligonucleotide that encodes the UTR sequence. During the first step of the 2-step cloning procedure, this oligonucleotide is ligated into the vector using two unique restriction sites & enzymes. The oligonucleotide contains within it a pair of unique restriction sites that were then used in the second cloning step to introduce the constant sfGFP cassette. This cloning step shifted the location of the barcode to the 3' UTR, but it does not alter the UTR's sequence or the barcode's sequence. The oligonucleotides are synthesized with an error rate of about 1 in 5000. PCR amplification also used a very high-fidelity DNA polymerase (NEB Q5) and a low number of cycles (22-25) to greatly minimize the introduction of additional mutations after synthesis. Thus, when we sequenced the barcodes using MiSeq and NovaSeq NGS, we found a very small number of mutated barcode sequences (about 0.1%). The barcode sequences were designed to be mutually identifiable even if they contain a nucleotide mutation (a minimum pairwise Hamming distance of 2) so we are still able to map and count these barcode sequences. We expect the UTR sequences to have a similar error rate (about 0.4% of UTR sequences with 1-nt errors, based on the UTR sequences being 4-fold longer than the barcodes, on average). Overall, we expect about 200 UTR sequences with a 1-nt mutation across the final train-test dataset of 45283 sequences.

The methods for the next-generation DNA and RNA sequencing and analysis are too brief to be reproduced by others. As described, “Amplicon libraries were then characterized using deep sequencing (Illumina NovaSeq, Genewiz).” I assume that the subsequent data

analysis uses only the perfect match reads as the “mapped” reads. However, it is not stated in the methods.

Yes, we’ve expanded the methods section to describe the library preparation and next-generation sequencing in greater detail, including data analysis. For additional methodological information, step-by-step protocols for these tasks are widely available in publications as well as within instruction manuals. Regarding the barcode mapping, we designed the barcodes to have pairwise Hamming distances of 2 so that they could be uniquely mapped even if they contained a 1-nt mutation. Over 99.9% of barcodes had perfect matches, while fewer than 0.1% contained a mutation.

It is not clear from the Methods if all kinetic measurements were performed in *E. coli* DH5-alpha. If that is correct, it would be good to briefly justify choosing this strain and discuss what effects it may have on using the model for other *E. coli* strains.

All of the *in vivo* measurements were carried out in *E. coli* DH5-alpha cells. While the enzymes & complexes responsible for mRNA degradation have been identified in *E. coli*, how their biophysical interactions with mRNA control degradation rates has not yet been quantified. Our work is the first to systematically vary these biophysical interactions (one at a time in some datasets, and in many combinations in other datasets, all measured at the same time in one massively parallel assay). However, these biophysical interactions and their effect on mRNA decay are not unique to *E. coli* DH5-alpha, but are likely well-conserved across many bacterial species. The results from this work are foundational and form the baseline for comparisons in other bacteria. We have extended our Discussion section to make these points.

Were the kinetic measurement assays performed more than once to test reproducibility of the reported data and reported mRNA degradation kinetic constant for each UTR design? It seems that the model would be less accurate if the dataset is a single experiment.

The kinetic measurements were determined from exponential decay curves using five time-pointed samples where each sample is cryogenically frozen (liquid nitrogen) at the reported times, followed altogether by RNA extraction, cDNA synthesis, and PCR amplicon generation. When analyzing the next-generation sequencing data, we set a high threshold for read depth to ensure that each DNA and cDNA sample were sequenced at least 100 times, which improves the precision of the time-pointed mRNA level measurements. We then compared the time-pointed mRNA level measurements to an exponential decay curve and only counted datapoints if all five time points exhibited exponential decay dynamics with a single mRNA half-life. There were 50048 mRNAs that exhibited purely exponential decay. The overall material cost of this experiment was about \$55,000 USD, which included several NovaSeq next-generation sequencing runs. Unfortunately, the level of funding for this project was insufficient to carry out another biological replicate, and we currently do not have available funds to carry out another biological replicate. The characterized biophysical interactions and design rules extracted from the dataset are a culminating pattern found in many thousands of datapoints, each subjected to experimental measurement noise. The model’s predictions were rigorously

tested on thousands of measurements not used to train the model (the unseen test dataset).

For the reported conclusions relating mRNA features to degradation rate (Figure 2B–2G), the statistical analyses to justify the conclusions seem to be missing. Another related point for the statistical analysis is that there is not a description or section in the Methods for the statistical analyses that were performed.

We've updated the manuscript text to additionally report the results of statistical analyses of the relationships shown in Figures 2B–G, supporting our observation that multiple sequence determinants collectively control mRNA decay rates and that 1-factor relationships can only be generally found if holding other factors relatively constant. For example, we found that longer single-stranded RNA lengths increase mRNA decay rate (up to about 30 nt long), but only when using a RppH binding site (first four nucleotides of the transcript) that confers higher mRNA stability and when the mRNA's translation initiation rate is relatively high (Figure 2C, Pearson $R = 0.924$, $p\text{-value} = 1.25\text{E-}14$, $N = 31$). The size effect is an increase of 0.01 1/minute in mRNA decay rate per single-stranded RNA nucleotide. In contrast, when the mRNA has a low translation initiation rate, the single-stranded RNA length (up to 30 nt) had little to no size effect (0.0008) on the mRNA's decay rate with a statistically significant comparison (Figure 2B, Pearson $R = -0.19$, $p\text{-value} = 1.62\text{E-}14$, $N = 30$).

We also found that the introduction of a G-quadruplex into the 5' untranslated region protects the mRNA from degradation. In the absence of a G-quadruplex, longer single-stranded RNA regions increase the mRNA's decay rate (up to 33 nt) with a statistically significant size effect of 0.01 1/min per nucleotide of ssRNA (Figure 2F, blue dots, Pearson $R = 0.926$, $p\text{-value} = 5.14\text{E-}15$, $N = 31$). In contrast, when designed G-quadruplexes of varying length are introduced into the 5' untranslated region, the mRNA's decay rates were all quite low (around 0.2 1/minute) and the equivalent length of the single-stranded RNA region – if the G-quadruplex did not form – had little to no size effect on the mRNA's decay rate with a statistically significant comparison (Figure 2F, orange dots, Pearson $R = 0.189$, $p\text{-value} = 8.1\text{E-}16$, $N = 31$). Finally, the introduction of the i-motif into the untranslated region was found to not have a protective effect. The measured mRNA decay rates were higher than an equivalent untranslated rate with a similar length (Figure 2G, $p\text{-value} = 4.13\text{E-}7$, $N = 27$). All statistical tests use a two-tailed Student's T-test (equal variances) with an alpha of 0.05.

The model testing datasets were subsets (20%) of the designed mRNA library with a random split of the mRNAs in each design group. However, the set of mRNA designs contained many highly similar UTR sequences. I would expect that the model would have accurate predictions if it were trained by highly similar sequences. Were the models used to predict the mRNA degradation rate of other transcripts that differ more greatly, such as those from other papers containing different promoters or CDSs, as a more generic test?

The mRNA sequences in each design subgroup were designed to systematically vary the strength of one, two, or three biophysical interactions, while keeping other interactions

the same. The model test dataset contains 20% of sequences from each design subgroup (randomly selected) to ensure uniformity of model testing across all design subgroups. Altogether, because there are so many design subgroups and so many sequences within each subgroup, the mRNA sequences within the model test dataset actually share very little sequence homology (fewer than 0.3 bits of information per aligned position).

Regarding cross-publication data analysis, prior experimental studies were not designed to measure kinetic mRNA decay rates while keeping other factors constant. For example, prior studies have measured mRNA levels while changing promoter sequences or measured fluorescent protein levels while changing 5' untranslated regions. A few past efforts have measured the decay kinetics of natural mRNAs in a transcriptome, but these mRNAs are also regulated at several additional levels (transcriptional, post-transcriptional), which greatly influences their stabilities. For a proper apples-to-apples comparison, it would be necessary to create an integrated multiple layered model of gene expression, which is a topic of a future study. The work here is focused on mRNA decay and the biophysical interactions that control its rate.

It was concluded that using only the single most predominant mRNA isoform lowered the model's accuracy of the steady-state mRNA level. However, if I am interpreting correctly, the supporting data analysis is a decrease of the testing dataset R-squared from 0.69 to 0.67. It should be discussed whether this is a significant change. I assume that if you randomly split the data multiple times, there would be some variation of these values.

Yes, it is correct that the 1-isoform version of the model still retains very high accuracy. We carried out a more detailed cross-model analysis to examine the differences in accuracy between the 1-isoform model and 5-isoform model and to see if they are statistically significant. Using the unseen test dataset, we calculated the prediction error for each model in terms of a fold-change (a ratiometric error) and found that the mean and median error for the 1-isoform model is higher than for the 5-isoform model. The mean fold-change error for the 1-isoform model is 1.584, while it's 1.563 for the 5-isoform model. The median fold-change error is 1.361 for the 1-isoform model, while it's 1.345 for the 5-isoform model. We carried out a two-tailed T-test (paired) to determine if the fold-change errors from the two models are statistically significant and found a p-value of $1.36E-7$ ($N = 9064$, $\alpha = 0.05$). The 5-isoform model does generate improved predictions with a statistically significant increase in accuracy (lower fold-change errors), but the size effect is relatively small. We have modified the manuscript text to include this statistical analysis in our results section.

Beyond relying solely on statistical analysis, we conclusively show that the first four nucleotides of the mRNA transcript have a large effect on the mRNA's stability (over a 4-fold effect). We also know that, when RNA polymerase initiates transcription, there is a stochastic kinetic "choice" for where to begin mRNA synthesis, depending on the discriminator and initial transcribed region sequences. The mechanism here is that the RNA polymerase – DNA complex will only transition from the open conformation to the processive stable conformation once the formation of a stable R-loop is complete (the R-loop is the DNA-RNA-DNA sandwich that stabilizes the melting of the double-stranded

DNA so that the template DNA can be read processively). Over the course of multiple initiation events, the stochasticity of this mechanism results in slightly different transcriptional initiation start sites, producing a mixture of mRNA isoforms where the first four nucleotides of the isoform can be shifted and distinct. Thus, from a mechanistic point-of-view, we have a strong hypothesis that promoters with more variable transcriptional start sites will produce more mRNA isoforms that can have different mRNA stabilities. We do indeed support this hypothesis with our data and analysis, even if the differences in model accuracy are relatively small when tested on this dataset.

It is important to retain and present the multi-isoform version of the model. First, future use of the model will include scenarios where the promoter sequence is quite different. There will be scenarios where there is more variation in the transcriptional start site locations and only the multi-isoform version of the model can predict these effects. We call this “future proofing” the model.

Second, it is generally unrecognized in the field that differences in transcriptional start sites could lead to differences in mRNA stabilities and gene expression levels. Therefore, it is important to show these differences even if they only affect a minority of mRNA sequences in our very large dataset. Future datasets could be designed to specifically investigate this effect, but only if it’s well-recognized as important in the field. There are also natural systems that use tandem overlapping promoters to produce many mRNA isoforms. Our results shed light on how those mRNA isoforms will have distinct mRNA stabilities.

The reported non-linear relationship between the mRNA degradation rate and the inverse of the steady state mRNA concentration (reported in the Discussion and Supplementary Figure 8) is interesting, yet this conclusion is not supported by statistical analysis.

We carried out a statistical analysis and confirmed that the relationship between the mRNA degradation rate and the inverse of the steady state mRNA concentration is not linear ($R = 0.04$). This analysis is included in the Supplementary Figure 8 legend.

The Methods do not seem to mention how the measured steady-state mRNA levels used for the model training and testing in Figure 3B were determined and whether they are directly from the experimental data. The reported units for mRNA concentration are arbitrary units (au), though it is not clear how these quantities were determined.

The methods section on “Data Analysis and Model Development” describes how we used the DNA-Seq, RNA-Seq, and spike-in control read counts (the raw experimental data) to determine the mRNA levels at each timepoint. The mRNA levels are reported in arbitrary units because all next-generation sequencing data only provides a proportional number on a relative scale (e.g. TPMs or RPKMs). Importantly, even though the mRNA levels are measured on a proportional relative scale, we can still extract exponential decay constants in units of real time (minutes). The change in state depends on a fold-reduction where the units cancel out [e.g. a half-life is the amount of time needed for an amount to be reduced by 2-fold, which is unitless].

The methods section on “Data Analysis and Model Development” reads: mRNA decay rates were determined from the DNA-Seq read counts, RNA-Seq read counts, and spike-in control RNA read counts at the 0, 2, 4, 8, and 16 minute post-rifampicin timepoints. RNA-Seq read count ratios (R_i) were calculated according to $R_i = N_i / N_0 / CR_i$, where N_i are the read counts at each timepoint, N_0 are the read counts at the 0 timepoint, and CR_i are the spike-in read count ratios at each timepoint. CR_i is calculated using $CR_i = C_i / C_0$, where C_i are the spike-in read counts at each timepoint and C_0 is the spike-in read count at the 0 timepoint. We then fit the time series R_0, R_2, R_4, R_8 , and R_{16} to an exponential decay curve with a single unknown parameter (k). We use non-linear least squares regression to identify the best-fit value of k , using the SciPy `curve_fit` function. We quantify the goodness of fit using Pearson R^2 . The source code for this analysis is found in **Supplementary Information**.

In line 475, the phrase “highly polymeric” is used to describe the subset of mRNAs. I am not familiar with what this means, and perhaps instead you mean that they contain homopolymeric regions or tracts. This also comes up in line 493.

Yes, that is correct. We’ve changed the manuscript text to read “contains homopolymeric regions” versus “highly polymeric”.

I did not find the mRNA Stability Calculator web interface available at the website provided. I assume you may be waiting until publication to release it.

Yes, the web interface for the mRNA Stability Calculator will be made available soon. We are combining multiple model predictions together and developing an improved interface so that researchers can visualize transcription rates, translation rates, and mRNA stabilities altogether.

The Introduction would be improved by expanding the discussion of bacterial mRNA degradation models that have been developed by other researchers.

We’ve expanded the Introduction to include additional mechanistic models, although these models were never experimentally tested on large datasets or have the ability to carry out sequence-to-function predictions.

I verified that the python script and test files are at the provided GitHub site. I did not run them. There is not a README file in the GitHub folder.

A README file has been added to the GitHub source code repository (top-level folder).

Reviewer 2

This study carries out a systematic analysis of mRNA stability in *E. coli* by coupling high-

throughput data collection with machine learning. In brief, the authors used massively parallel sequencing to measure mRNA-specific degradation rates for a large library of carefully designed, sequence-diverse mRNAs (62,120), employed rational multi-factor analyses and biophysics-guided machine learning methods (i.e., a gradient-boosting framework) to examine the molecular determinants of mRNA stability within their library, and developed predictive algorithms to estimate time-dependent mRNA levels (i.e., mRNA levels at specific time points post inhibitor-mediated transcriptional arrest) and degradation kinetics. Overall, the large mRNA library, which includes half-lives that vary by 10- to 100-fold, was sufficiently variable to permit an assessment of the isolated influence—or lack thereof—of important design factors (e.g., translation rates, secondary structures, sequence compositions, G-quadruplexes, i-motifs, and RppH activity); it also enabled the development of models with reasonable predictive power. The paper is well-written, easy to follow, and may be appropriate for Nature Communications. In its present form, however, the study makes observations and conclusions that are lacking in statistical significance (i.e., they are based on plots where differences between data points are not clear and/or lack a statistical test), and the generalizability of the models across genetic contexts and growth conditions is not assessed. These weaknesses can probably be remedied.

We thank the reviewer for their positive feedback and have sought to address their concerns with additional statistical analysis and explanations, described below.

1. Line 97: protein coding sequence (CDS). The entire paper is based on one CDS (GFP), which is understandable. The CDS, however, can affect mRNA stability—at least in human cells (see reference). Of course, repeating the entire analysis with a different protein is beyond the scope of this study, but because the study already identifies a range of mRNA stabilities, it would be informative to compare a subset of mRNAs (e.g., 10) with a few different proteins (e.g., GFP, maltose binding proteins, perhaps an inactive enzyme). Are the mRNA half-lives the same, regardless of CDS?

Ref: Nature Communications volume 13, Article number: 6829 (2022)

Yes, as we show in our work, we use our RBS Calculator model to predict the translation initiation rate of the CDS, which is then used in our mRNA Stability Calculator to predict the mRNA's decay rate. Using our large dataset, we show that these predicted translation rates are an important determinant of the mRNA's stability by changing the number of bound ribosomes that are protecting the mRNA from RNase activity. Previously, the RBS Calculator model has been experimentally validated using many CDS sequences (see Reis & Salis, ACS Synthetic Biology, 2020) and is now widely used in the field to control the translation rates of diverse CDSs (enzymes, transcription factors, structural proteins, etc). Thus, the current model has the ability to predict the mRNA stability of transcripts that express very different CDS sequences, explicitly taking into account how the CDS sequence controls its translation rate. While there may be other interactions (encoded in the CDS sequence) that can affect the mRNA's stability, we would need to carry out a massively parallel assay on many designed sequences (like we performed here) to systematically investigate and characterize those interactions. Unfortunately, adding 10 more datapoints to the dataset would not be sufficient to report any new claims about

those interactions. We do indeed have plans to characterize a large library of genetic systems where the CDSs have been designed to systematically vary key parameters that control their gene expression interactions (using a new oligopool-to-plasmid build workflow that makes it substantially cheaper to build such libraries).

2. Related to the above. It seems logical that growth conditions, specifically those that might elicit some form of cellular stress, could re-weight the relative contributions of different mRNA attributes (e.g., RppH activity). It would be informative to take a subset of mRNAs with different half-lives (e.g., 10) and, ideally, different half-life-affecting features, and evaluate them in a new growth condition. Do the half-lives change between growth conditions? Do they correlate between conditions? This analysis, like the one above, would go a long way in establishing generalizability. Importantly, even if the results are not generalizable—across CDSs or growth conditions—the findings would be valuable and increase the impact of the study.

Ref: Front Microbiol. 2020; 11: 2111. Published online 2020 Sep 9. doi: 10.3389/fmicb.2020.02111

Yes, it is well-known that stress-induced conditions can affect key variables inside the cell that will have a global effect on the stabilities of all mRNAs at the same time. For example, during the stationary growth phase, there are considerable changes to the number of available ribosomes and their ability to initiate translation and protect mRNAs from RNase activity, which will increase or decrease the stability of all mRNAs inside the cell at the same time. However, interestingly, the expression of many of the enzymes that participate in RNA processing and degradation are regulated by their own feedback loops that confer homeostasis, even in response to perturbative conditions. For example, RNase E expression is self-regulating; excess amounts of RNase E will destabilize the *rne* transcript [Mudd & Higgins 1993]. Previous experiments have found that over-expression of RppH did not alter mRNA decay rates in a sequence-specific manner [Luciano et. al. *J. of Bacteriology* 2012].

In our work, we focused on characterizing the sequence-determining biophysical interactions that control a mRNA's stability and degradation rate. We identified and quantified several interactions via a combination of rational sequence design, massively parallel kinetic measurements, and biophysics-guided machine learning. We tested our model's mRNA decay rate predictions on over 9000 datapoints in the unseen test dataset. Researchers can input arbitrary mRNA sequences into our model and obtain good mRNA stability predictions, accounting for all the listed biophysical interactions characterized in our work. The questions in our work are well-defined and answered. However, if we begin adding stress-induced conditions into the environmental variables, we will be engaged in a highly expanded effort that will require collection of many thousands of datapoints to arrive at a satisfactory & conclusive answer. While we agree that these are important questions, they deserve their own focused effort to answer them.

3. Line 99: complete paragraph: I found this paragraph and the following paragraph to be somewhat confusing. What is being held constant and what is being changed within each design group? Could the authors clarify further? The wording is a little confusing.

Yes, in each design group, we designed a large number of mRNA sequences to systematically vary either one, two, or three investigated biophysical properties in combinations, while keeping the other investigated biophysical properties the same. The properties being varied in each design group are specific to that design group. For example, in one design group, there are 1280 mRNA sequences where each sequence has varied RppH binding sites (256 variants) and varied ribosome binding sites (5 variants) in combination with an intervening polyA single-stranded RNA region. Overall, there are 23 design groups that systematically explore the sequence-structure-function space. In the Supplementary Data 2, we list all sequences, their design group, and the biophysical properties being calculated for each sequence.

4. Line 231: “we find a clear inverse relationship between the CDSs’ predicted translation initiation rates and their measured decay rates (Pearson $R = -0.31$).” This trend does not look significant. At most, it is a very weak correlation. Could the authors comment on the significance of the trends?

Yes, the presented 1-factor relationship between the mRNA’s predicted translation initiation and its measured decay rate has a weak correlation (Pearson $R = -0.31$), though it is statistically significant (two-tailed T-test, equal variances, p -value = $4.22E-27$). An important point made in this result section “Multi-Factor Sequence Determinants Controlling mRNA Decay” is that multiple factors work together to control a mRNA’s decay rate because mRNA degradation is a complex multi-step process where multiple biophysical interactions collectively control where and how often the first cleavage event takes place. Therefore, when plotting a single predicted factor vs. the measured mRNA decay rate, we only observe weak correlations for individual factors. However, when we combine all the predicted biophysical properties together and use machine learning (LightGBM) to learn a (non-linear) relationship, we develop and validate a much more accurate model. We have modified the figure legend to include this additional statistical analysis.

5. Line 245: “PolyA motifs appeared to have a larger impact on mRNA decay than other polymeric motifs, particularly when mRNAs had high predicted translation rates (Figure 2E).” How so? The mRNA decay rates for mRNAs with polyA motifs were highly variable (more so than for polyG). Wouldn’t that imply that polyA tails do not strongly impact decay rates?

Yes, the variability in decay rate within the polyA single-stranded RNA categories is large, but the mean decay rates are higher than all categories of single-stranded RNA regions. The higher mean decay rates are an indicative that polyA single-stranded RNA is more accessible and/or more likely to bind to RNases. The high variabilities are an indication that other biophysical properties are confounding this 1-factor effect, for example, when mRNAs have less active RppH binding sites that slow down mRNA decay.

6. Line 248: “Finally, we found that G-quadruplex tertiary structures had a strong protected effect on mRNA stability, lowering their decay rates, while the i-motif tertiary

structures had not apparent protective effect on mRNA stability (Figure 2FG).” This statement is not supported by the data. In Figs. 2F-2G the differences between the blue and orange points do not appear to be statistically significant. Could the authors justify their statement with a statistical test. Perhaps, they could find a window where it is true (e.g., ssRNA length of 30-40 for G-quadruplex).

The best way to interpret these results is that the introduction of a G-quadruplex stabilized the mRNA’s stability (maintained a low decay rate) as compared to an equivalent length of mRNA region where the mRNA region would otherwise have formed single-stranded RNA. A mRNA region with no single-stranded RNA will be stable (with or without a G-quadruplex). A mRNA region with 30 nucleotides of single-stranded RNA will be very unstable. But if we replace those 30 nucleotides of single-stranded RNA with a sequence that forms a G-quadruplex (this only requires a few mutations), then the mRNA decay decreases quite a lot. We can apply statistics to compare the size effects of these relationships. In the absence of a G-quadruplex, longer single-stranded RNA regions increase the mRNA’s decay rate (up to 33 nt) with a statistically significant size effect of 0.01 1/min per nucleotide of ssRNA (Figure 2F, blue dots, Pearson R = 0.926, p-value = 5.14E-15, N = 31). In contrast, when designed G-quadruplexes of varying length are introduced into the 5’ untranslated region, the mRNA’s decay rates were all quite low (around 0.2 1/minute) and the equivalent length of the single-stranded RNA region – if the G-quadruplex did not form – had little to no size effect on the mRNA’s decay rate with a statistically significant comparison (Figure 2F, orange dots, Pearson R = 0.189, p-value = 8.1E-16, N = 31).

We ran the same analysis on the i-motif, but found that the introduction of the i-motif into the untranslated region was found to not have a protective effect. The measured mRNA decay rates were higher than an equivalent untranslated rate with a similar length (Figure 2G, p-value = 4.13E-7, N = 27). All statistical tests use a two-tailed Student’s T-test (equal variances) with an alpha of 0.05.

7. Lines 260-263: same comment as above. Please check the statistical significance of the observations made about Figs. 2F-2G.

Yes, we ran statistical analysis on all relationships shown in Figure 2. These calculations are now all reported in the manuscript text.

8. Line 383: “found that RppH binding sites with polymeric compositions (skewed statistics toward one nucleotide) greatly lowered mRNA stabilities as compared to RppH binding sites with balanced compositions (Figure 4A).” Based on the data, could it be equally likely that A-rich binding sites improve stability? The least stable binding sites are A poor (i.e., there are no polymeric A binding sites in this set).

We initially had the same thought, but when examining all the RppH binding site effects (Supplementary Figure 7), the AAAA motif has the 19th lowest mRNA level (out of 256). The least stable RppH sites also include TTTT, GGGG, CCCC, which suggested to us that homopolymeric sequences were more likely to bind to RppH and lower mRNA levels.

9. Line 394: “The model also quantitatively shows that longer single-stranded RNA regions in the 5’ UTR lead to lower mRNA stability, although the effect is most pronounced for C-rich and A-rich ssRNA sequences (Figure 3B).” Both statements seem to lack statistical significance. The impact of the single-stranded region seems extremely modest (i.e., nearly a flat line) and the differences between polyA and polyU or polyG seem insignificant. PolyC is a bit more influential, but is it significant? Can the authors use a statistical test to substantiate the importance of the trends that they are calling out? A reference that would justify difference in stability as impactful/not impactful, from a biological perspective, could also help.

These statements are the outcome of the machine learning analysis (LightGBM, a gradient boosted decision tree), which is itself an advanced statistical test. The machine learning procedure uses the training dataset (36219 measurements on diverse mRNAs) to identify the most important biophysical properties and their contributions to the measured mRNA decay rates in order to develop a sequence-to-function quantitative model. This model was then tested on the unseen test dataset (9064 measurements on diverse mRNAs) to ensure that its predictions are accurate and generalized across these designed mRNAs. We then ask the model “What were the effects of these biophysical properties on the mRNAs’ decay rates?”. The plots in Figure 4 show those answers.

Importantly, LightGBM excludes data outliers and only assigns an importance to a feature (a biophysical property) if many datapoints support the importance of that feature. It will also quantify the importance of the feature in a way that best generalizes across all the training dataset. By its very construction and purpose, when a LightGBM model predicts a change in outcome when a feature input is changed, it has already learned (from many datapoints) that there exists a statistically significant relationship between the feature input and the outcome. Relatedly, we selected the LightGBM model type because its architecture and training/testing workflow are highly resistant to overfitting to the dataset and “memorizing” outcomes. For example, the LightGBM training workflow minimizes the number of decisions (“splits”) and contributions (“leaves”) if they are not directly supported by the dataset. Altogether, this procedure has a “smoothing” effect so we do not expect to see very large differences in the model predictions when varying a single biophysical property. For the same reasons, the dynamic range in the model predictions is also smaller than the dynamic range of the measurements.

10. Line 421: “Over 80% of the characterized UTR variants exhibited purely exponential mRNA decay.” What test are the authors using to compare models? Did they compare different kinetic equations for decay?

We used non-linear least squared regression to quantify how well each mRNA level time-course fit to an exponential decay function with a single half-life. Over 80% of the mRNA level time-course data fit well to such an exponential decay function with a Pearson R^2 of 0.75 or greater. Many of the remaining mRNA level time-course data had either flat or highly volatile mRNA levels. All data is available in our Supplementary Data files and can be further studied in future efforts.

11. Line 491: The above point comes up again as one of the major conclusions of the paper. “Polymeric C-rich and A-rich single-stranded RNA regions lowered mRNA stability more so than U-rich or G-rich sequences, suggesting that sequence composition has a direct impact on the persistence length....” These differences do not seem significant.

These relationships were found to be statistically significant during the training, testing, and cross-validation of the LightGBM model (as part of the training and testing workflow). However, it is true that when looking at Figure 3B, the size effect of changing just the sequence composition of the single-stranded RNA region can be small. For example, the LightGBM model predicts that changing a 40-nucleotide region from primarily polyG composition to polyC composition will reduce the mRNA level by about 1.5-fold. But, in reality, when we change a mRNA’s sequence, we will alter multiple biophysical properties at the same time. The same change in sequence composition to this 40-nt region could also lower the translation rate of the mRNA by 100-fold, which will further lower the mRNA’s level by 2-fold. The individual effects from these contributions are added together and now the total change in mRNA level is 3-fold. If these mutations to the mRNA also included the first 4 nucleotides of the transcript, then we could see another 4-fold decrease in the mRNA level and so the final tally might be a 12-fold decrease in mRNA level. Overall, if we scrutinize the contributions of individual biophysical properties, they might appear small. But, when we tally up the total contributions from all the biophysical properties, then the final change in the outcome can be large.

Minor Points

1. Line 84: Fig. 1C. This plot lacks a legend. To what do the different colors correspond? How were the relevant samples chosen? Are there estimates of error (for points or fits)?

Figure 1C shows how four different mRNA level time-course data curves were fit to the exponential decay function. The dots are experimental measurements (RNA-Seq read counts divided by DNA-Seq read counts, using the Spike-in controls for normalization). The lines are the fitted exponential decay curves. The four time-course data curves were selected to illustrate the range of mRNA half-lives observed across the measurement dataset. Figure 1D shows the measured mRNA half-lives for all time-course data curves.

2. Line 153: How did the authors pick the dosing level / concentration of Rifampicin? I ask only because a high concentration could elevate cellular stress (which may matter).

500 ng/uL rifampicin was selected as this concentration is sufficiently high to completely halt transcriptional initiation for all RNAP inside *E. coli* cells. This is the same concentration of rifampicin used in prior work that measured the mRNA decay rates of natural mRNAs [<https://www.pnas.org/doi/full/10.1073/pnas.112318199>].

Predicting synthetic mRNA stability using massively parallel kinetic measurements, biophysical modeling, and machine learning

Daniel P. Cetnar¹, Ayaan Hossain², Grace E. Vezeau³, and Howard M. Salis^{1,3,4*}

NCOMMS-24-01006

We would like to thank the reviewers again for providing additional constructive comments. Below, we describe the additional text changes to our manuscript as well as the additional README file that should address the reviewers' requests. The reviewers' requests are listed below, followed by our responses (**green**). We've included a version of the manuscript with changes highlighted (in the peer review package).

Reviewer #1:

The added README file does not provide instructions for installing and running the script in this project. It only includes the abstracts and citations for six different papers. There should be a README within the project folder and ideally it will contain requirements for the script and brief instructions.

We have added a README file in the project folder that lists the Python dependencies needed. Code instructions are provided to repeat the development and testing of the mRNA Stability Calculator model.

Reviewer #2:

The reviewers have done a nice job responding to my questions; clarifying changes to the text, however, were minimal—a perplexing decision, given the thoughtful replies. The text would be clearer, and the analysis, more rigorous, if the authors incorporated elements from their responses into the actual paper. In general, there is a significant disconnect between the data and the words used to describe the data; the text still tends toward a blunt, overly broad interpretation of extremely weak trends: There are several key opportunities for quick remedies:

We are delighted to include additional analysis (from our previous response) into the manuscript text. When revising the manuscript text, we have taken care to avoid qualitative subjective adjectives to describe trends in the data. Instead, we point to the relevant metrics to quantify the importance of each interaction on the outcome (ie, letting the data speak for itself).

1. Responses to prior comments 6 and 7, which relate to Figures 2F and 2G. The authors claim that they “ran statistical analyses on all relationships shown in Figure 2” and that “these calculations are now all reported in the manuscript text”, but they are not—not for 2F and 2G, where the relationships are suspect.

Questionable statements from text:

“Finally, we found that G-quadruplex tertiary structures had a strong protective effect on mRNA stability, lowering their decay rates, while i-motif tertiary structures had no apparent protective effect on mRNA stability (Figure 2FG).”

“We found that the introduction of G-quadruplexes into the 5' UTR lowered the mRNAs' decay rates (to a constant of about 0.2 1/min) regardless of the secondary mRNA structures that would otherwise form with varied amounts of ssRNA (Figure 2F). In contrast, using the same analysis, we found that the introduction of i-motifs could greatly increase mRNA decay rates, compared to an equivalent mRNA that only formed secondary mRNA structures (Figure 2G).”

Comment: “Strong protective effect” is an absurd overstatement. Look at the plots. Again, the authors mention a statistical test, but they do not actually include that test in the figure caption. Also, it is not clear how they are calculating the two-tailed T Test, which is mentioned only in their response, as it relates to this data, not in the actual test. It is OK to say “weak, but statistically significant protective effect for intermediate lengths”. Let the data speak.

We modified the sentence “we found that G-quadruplex tertiary structures had a strong protective effect on mRNA stability” to remove the word “strong”. We also added the comparison of size effects to justify our conclusions. This paragraph has the addition:

“In the absence of a G-quadruplex, a longer ssRNA region caused the mRNA decay rate to increase by about 0.01 1/minute per ssRNA nucleotide up to a length of about 33 nucleotides (linear regression; $R = 0.926$, $p = 5.14 \times 10^{-15}$, $N = 31$) with a peak mRNA decay rate of about 0.4 1/min. But when a G-quadruplex was introduced into the 5' UTR, the mRNA decay rate was generally constant at about 0.2 1/min and did not depend on the 5' UTR length (for up to 33 nucleotides of added sequence).”

Likewise, we explicitly describe the effect of adding an i-motif to a 5' UTR by adding the sentence “For example, introducing an i-motif into a shorter 5' UTR caused its mRNA decay rate to be high (about 0.5 1/min) in comparison to mRNAs with an equivalent length of ssRNA (5 to 10 nt), which had mRNA decay rates of about 0.2 1/min.”

These sentences succinctly describe the effect of adding either a G-quadruplex or an i-motif to a 5' UTR, using the quantitative measurements to indicate the degree of change rather than a subjective word. It would be incorrect to describe these effects as “weak” as a 2-fold decrease in mRNA decay rate from 0.4 1/min to 0.2 1/min is not a weak effect. It is more correct to describe specific scenarios of what happens when adding a G-quadruplex or i-motif to a 5' UTR, using specific datapoints and measurements to quantitatively pinpoint the outcome. In this way, the reader can better understand each interaction and use these illustrative examples as a guide towards designing their own synthetic mRNAs.

2. Responses to comments 4 and 5. As above, the main text statements still overstate the trends. Look at the data—it doesn't match the description. The trends are still extremely weak. It would be clearer—and, notably, more informative—to say this: “We see a very weak, inverse relationship between...”. For 2E, it seems like the variability in the effect of the PolyA motifs is the more interesting result. The mean decay rates may be higher, but it does not look statistically significant, and the statistical tests, again, are not present in the relevant caption E.

Questionable statements from text:

“We find a clear inverse relationship between the CDSs' predicted translation initiation rates and their measured decay rates (Pearson $R = -0.31$, $N = 1527$ mRNAs) with higher translation rates providing mRNAs with more stability (Figure 2A), confirming that some design factors may only have a clear effect on mRNA decay rates when other contributing factors are kept relatively constant.”

Again, following a philosophy of generally letting the data speak and avoiding any subjective terminology, we have modified the first sentence (changes bolded) to:

“Notably, when we focus on mRNAs with stable RppH binding sites and moderate ssRNA lengths, we find an **clear** inverse relationship between the CDSs' predicted translation initiation rates and their measured decay rates (Pearson $R = -0.31$, **p-value = 4×10^{-27}** , $N = 1527$ mRNAs) with higher translation rates providing mRNAs with more stability (**Figure 2A**), confirming that some design factors may only have a clear size effect on mRNA decay rates when other contributing factors are kept relatively constant.”

A Pearson correlation coefficient of -0.31 correctly quantifies an inverse relationship between an independent variable (predicted translation initiation rate) and a dependent variable (measured mRNA decay rate). It is statistically significant as quantified by the very small p-value. The importance of this correlation depends on the system of interest. In our system, there are several interactions that affect a mRNA's decay rate, but this plot only shows the contribution of one interaction (translation initiation rate) on the outcome. We do not expect to see a high Pearson correlation coefficient when using only one interaction to “predict” the outcome (importantly, our analysis does not claim to predict the outcome from this interaction alone). Instead, as we show with the machine learning analysis, we need to take into account multiple interactions to predict the outcome with higher Pearson correlation coefficients (unseen test dataset $R = 0.83$ for steady-state mRNA levels and unseen test dataset $R = 0.66$ for mRNA decay rates).

Our manuscript explicitly leads the reader down a road of cataloguing the important interactions that affect mRNA decay (by some amount) and then applying machine learning to combine all the interactions' effects to predict the outcome. You can see that when we introduce the “Multi-Factor Sequence Determinants Controlling mRNA Decay” section by writing:

“Our first data analysis step was to determine how the sequence design factors affected the measured mRNA decay rates. Initially, we compared how changing a single design factor altered mRNA decay rates. However, we found that focusing on only a single design factor resulted in larger intra-category differences than inter-category differences, indicating that multiple factors are collectively controlling the mRNA’s decay rate with equally important contributions.”

We properly state the expectation that individual interactions only partly control mRNA decay rates, requiring a more sophisticated analysis that takes into account the effects of multiple interactions working together. We then carry out that multi-interaction analysis to arrive at the mRNA Stability Calculator model.

PolyA motifs appeared to have a larger impact on mRNA decay than other homopolymer motifs, particularly when mRNAs had high predicted translation rates (Figure 2E).

We have changed this manuscript text to:

“PolyA motifs appeared to have a larger impact on mRNA decay than other homopolymer motifs, particularly when mRNAs had high predicted translation rates (**Figure 2E**), **based on both the larger changes in the median mRNA decay rates and the greater spread in their distributions.**”

We would like to note that Figure 2D and Figure 2E show data in **boxplots**. To clarify, box plots show the median value of each category (the orange line), the 25% and 75% quartile boundaries (the boxes), the maximum and minimum of the in-distribution data (the bars), and the outlier data (the individual points). The bars in Figure 2DE are not error bars and they are not related to the standard deviation. The bars in Figure 2DE are the highest and lowest measurement values that could have been sampled from the datapoints’ apparent probability distribution. The bars are the maximum range of the data (excluding the small number of datapoint outliers that statistically fall outside the apparent distribution). Thus, the bars should encompass a large swath of the measurement space. We have now included the boxplot symbology in the Figure 2 legend: “Box plots show the median value of each category (orange line), the 25% and 75% quartile boundaries (boxes), the maximum and minimum of the in-distribution data (bars), and the outlier data (points).”

The best way to visualize the data differences across categories is to compare the median and quartile values. We clearly see that the quartiles for the PolyA motifs are shifted upwards (higher mRNA decay rates) in Figure 2E as compared to other motifs. The shift is most apparent for longer PolyA motifs; the *lower* quartile for PolyA motifs (11-20 nt) is higher than the *upper* quartile for most other motifs, which is a big effect.

It is true that the spread in the PolyA motif data is larger than the other motifs, which we now note in the manuscript text. The explanation for this larger spread is noted later in the manuscript when discussing the model development, “The model also quantitatively shows that longer single-stranded RNA regions in the 5’ UTR lead to lower mRNA stability, although the effect is most pronounced for C-rich and A-rich ssRNA sequences. PolyA

nucleic acids are known for being particularly straight and stiff, which can increase their accessibility to RNase activity⁵⁶.

Review of Cetnar et al

The reviewers have done a nice job responding to my questions; clarifying changes to the text, however, were minimal—a perplexing decision, given the thoughtful replies. The text would be clearer, and the analysis, more rigorous, if the authors incorporated elements from their responses into the actual paper. In general, there is a significant disconnect between the data and the words used to describe the data; the text still tends toward a blunt, overly broad interpretation of extremely weak trends: There are several key opportunities for quick remedies:

1. Responses to prior comments 6 and 7, which relate to Figures 2F and 2G. The authors claim that they “ran statistical analyses on all relationships shown in Figure 2” and that “these calculations are now all reported in the manuscript text”, but they are not—not for 2F and 2G, where the relationships are suspect. See below.

(F) G-quadruplex tertiary structures or (G) i-motif tertiary structures into highly translated mRNAs as compared to highly translated mRNAs lacking each type of structure, each indexed by the amount of single-stranded RNA in their 5' UTRs.

Questionable statements from text:

- “Finally, we found that G-quadruplex tertiary structures **had a strong protective effect** on mRNA stability, lowering their decay rates, while i-motif tertiary structures had no apparent protective effect on mRNA stability (Figure 2FG).”
- “We found that the introduction of G-quadruplexes into the 5' UTR lowered the mRNAs' decay rates (to a constant of about 0.2 1/min) regardless of the secondary mRNA structures that would otherwise form with varied amounts of ssRNA (Figure 2F). **In contrast, using the same analysis, we found that the introduction of i-motifs could greatly increase mRNA decay rates, compared to an equivalent mRNA that only formed secondary mRNA structures (Figure 2G).**”

Comment: “Strong protective effect” is an absurd overstatement. Look at the plots. Again, the authors mention a statistical test, but they do not actually include that test in the figure caption. Also, it is not clear how they are calculating the two-tailed T Test, which is mentioned only in their response, as it relates to this data, not in the actual test. It is OK to say “weak, but statistically significant protective effect for intermediate lengths”. Let the data speak.

2. Responses to comments 4 and 5. As above, the main text statements still overstate the trends. See below:

Figure 2: Multi-Factor Sequence Determinants Controlling Bacterial mRNA Decay (A) Measured mRNA decay rates are shown when varying the predicted mRNA translation initiation rates, while only including 5' UTRs with stable (less active) RppH binding sites and moderate single-stranded RNA lengths (Pearson $R = -0.31$, $p = 4.2 \times 10^{-27}$). (B) Measured mRNA decay rates are shown when varying single-stranded RNA lengths inside the 5' UTR, while only including 5' UTRs with stable RppH binding sites and low predicted translation initiation rates (<5000 au) (small size effect = 0.0008 for 0 to 30 nt, Pearson $R = -0.19$, $p = 1.6 \times 10^{-14}$). (C) Measured mRNA decay rates are shown when varying single-stranded RNA lengths inside the 5' UTR, now including 5' UTRs with stable RppH binding sites and high predicted translation initiation rates (>5000 au) (large size effect = 0.01 for 0 to 30 nt, $R = 0.924$, $p = 1.3 \times 10^{-14}$). (D, E) Measured mRNA decay rates are shown when varying the length and composition of single-stranded RNA regions inside 5' UTRs, using ribosome binding sites with either (D) low or (E) high predicted translation initiation rates. (F,G) Measured mRNA decay rates are shown when introducing either

Questionable statements from text:

- “We find a clear inverse relationship between the CDSs’ predicted translation initiation rates and their measured decay rates (Pearson $R = -0.31$, $N = 1527$ mRNAs) with higher translation rates providing mRNAs with more stability (Figure 2A), confirming that some design factors may only have a clear effect on mRNA decay rates when other contributing factors are kept relatively constant.”
- **PolyA motifs appeared to have a larger impact on mRNA decay** than other homopolymer motifs, particularly when mRNAs had high predicted translation rates (Figure 2E).

Comment: Look at the data—it doesn’t match the description. The trends are still extremely weak. It would be clearer—and, notably, more informative—to say this: “We see a very weak, inverse relationship between...”. For 2E, it seems like the variability in the effect of the PolyA motifs is the more interesting result. The mean decay rates may be higher, but it does not look statistically significant, and the statistical tests, again, are not present in the relevant caption E.